# Genome-wide prediction of DNase I hypersensitivity using gene expression

Weiqiang Zhou [1], Ben Sherwood[1,3], Zhicheng Ji[1], Yingchao Xue[2], Fang Du[1], Jiawei Bai[1], Mingyao Ying[2] & Hongkai Ji [1]

We evaluate the feasibility of using a biological sample's transcriptome to predict its genome-wide regulatory element activities measured by DNase I hypersensitivity (DH). We develop BIRD, Big Data Regression for predicting DH, to handle this high-dimensional problem. Applying BIRD to the Encyclopedia of DNA Elements (ENCODE) data, we found that to a large extent gene expression predicts DH, and information useful for prediction is contained in the whole transcriptome rather than limited to a regulatory element's neighboring genes. We show applications of BIRD-predicted DH in predicting transcription factor-binding sites (TFBSs), turning publicly available gene expression samples in Gene Expression Omnibus (GEO) into a regulome database, predicting differential regulatory element activities, and facilitating regulome data analyses by serving as pseudo-replicates. Besides improving our understanding of the regulome–transcriptome relationship, this study suggests that transcriptome-based prediction can provide a useful new approach for regulome mapping.

[1] Department of Biostatistics, Johns Hopkins University Bloomberg School of Public Health, 615 North Wolfe Street, Baltimore, MD 21205, USA. [2] Department of Neurology, Hugo W. Moser Research Institute at Kennedy Krieger and Johns Hopkins University School of Medicine, Baltimore, MD 21205, USA. [3] Present address: School of Business, University of Kansas, 1654 Naismith Drive, Lawrence, KS 66045, USA. Correspondence and requests for materials should be addressed to H.J. (email: hji@jhu.edu)

A fundamental question in functional genomics is how genes' activities are controlled temporally and spatially. To answer this question, it is crucial to comprehensively map activities of all genomic regulatory elements (i.e., regulome) and understand the complex interplay between the regulome and transcriptome (i.e., transcriptional activities of all genes). Regulome mapping has been accelerated by high-throughput technologies such as chromatin immunoprecipitation coupled with high-throughput sequencing (ChIP-seq)[1] and sequencing of chromatin accessibility (e.g., DNase-seq[2] for DNase I hypersensitivity (DH), FAIRE-seq[3] for formaldehyde-assisted isolation of regulatory elements, and ATAC-seq[4] for assaying transposase-accessible chromatin). So far, these technologies have only been applied to interrogate a small subset of all possible biological contexts defined by different combinations of cell or tissue type, disease state, time, environmental stimuli, and other factors. A major limitation of these technologies is the difficulty in simultaneously analyzing a large number of different biological contexts. This limitation along with practical constraints such as lack of materials, antibodies, resources, or expertise has hindered their application by the vast majority of biomedical investigators from small laboratories.

Numerous researchers have examined how genes' transcriptional activities can be predicted using activities of their associated regulatory elements[5–7]. However, the interplay between regulome and transcriptome is bidirectional due to feedback[8, 9]. A systematic understanding of this relationship in the reverse direction, to what extent regulatory elements' activities can be predicted by transcriptome, is still lacking.

Here we investigate the problem of using gene expression to predict DH based on the data generated by the Encyclopedia of DNA Elements (ENCODE) Project[10]. Besides creating a more complete picture of the regulome–transcriptome relationship, this investigation also has important practical implications for regulome mapping. Gene expression is the most widely measured data type in high-throughput functional genomics. Measuring expression does not require large amounts of materials and complex protocols, and technologies for expression profiling are relatively mature. Thus, expression data are routinely collected when other functional genomic data types are difficult to generate due to technical or resource constraints. Today, the Gene Expression Omnibus (GEO) database[11] contains 200,000+ human gene expression samples from a broad spectrum of biological contexts, as compared to only ~7000 human ChIP-seq, DNase-seq, FAIRE-seq, and ATAC-seq samples. If one can use the ENCODE data to build prediction models and apply them to existing and new transcriptome data to predict regulome, the catalog of biological contexts with regulome information may be quickly expanded (Fig. 1a). This will provide a useful approach for regulome mapping that is complementary to existing experimental methods. It will also allow researchers to more effectively use expression data to study gene regulation. Unlike a recent study that imputes one functional genomic data type based on multiple other data types, which are non-trivial to collect[12], prediction in this study is based on one single but widely available data type and hence can have a substantially broader range of applications. During our investigation, we develop a big data regression approach, BIRD (Fig. 1b), to handle the prediction problem where both predictors (i.e., transcriptome) and responses (i.e., regulome) are ultra-high-dimensional, which is an emerging problem in the analysis of big data. We show that BIRD can provide practically useful predictions of chromatin accessibility using gene expression. BIRD-predicted DH can be used to predict transcription factor-binding sites (TFBSs), turn publicly available gene expression samples in GEO into a regulome database, and serve as pseudo-replicates to facilitate regulome

data analyses. It can also be used to predict differential regulatory element activities such as changes of chromatin accessibility between different cell types or differentiation time points.

## Results

**Big data regression for predicting DH.** Predicting DH using gene expression can be formulated as a problem of building millions of regression models, one per genomic locus, to describe the relationship between the DH level at that locus (response) and the expression levels of tens of thousands of genes (predictors). For method development and evaluation, we compiled DNase-seq and exon array (i.e., gene expression) data for 57 distinct human cell types with normal karyotype from ENCODE (Supplementary Data 1). They were randomly partitioned into a training dataset (40 cell types) and a test dataset (17 cell types). After filtering out genomic regions with weak or no DH signal across all 40 training cell types, 912,886 genomic loci (referred to as "DNase I hypersensitive sites" or "DHSs" hereinafter) with unambiguous DNase-seq signal in at least one training cell type were retained for subsequent analyses ("Methods").

For each locus, prediction models were constructed using the 40 training cell types. Prediction performance was evaluated using the 17 test cell types based on three types of statistics (Fig. 2a, "Methods"). First, we evaluated a method's ability to predict the variation of DH levels across different genomic loci by computing the Pearson's correlation between the predicted and true DH values (or P–T correlation) across all DHSs within the same cell type (cross-locus correlation $r_L$). Second, we evaluated a method's ability to predict the relative activities of the same DHS in different cell types by computing the P–T correlation across different cell types at each genomic locus (cross-cell-type correlation $r_C$). Third, we computed the total squared prediction error normalized by the total DH data variance ($\tau$) to characterize the proportion of data variation not explained by the prediction.

The regression for each locus can be constructed using either its neighboring genes or all genes as predictors (Supplementary Fig. 1). We tested both strategies (see "Methods" and Supplementary Figs. 2–6 for details). The latter strategy requires one to deal with a challenging big data regression problem, which involves fitting ~1 million high-dimensional regression models, each with a large number (18,000+) of predictors and small sample size. To cope with the high dimensionality and heavy computation, we developed the BIRD algorithm. BIRD predicts DH at each genomic locus by combining predictions from two types of models, locus-level model and pathway-level model, through model aggregation (Fig. 1b).

The locus-level model, denoted by $BIR(\overline{X}, Y)$, groups correlated predictors (i.e., co-expressed genes) into clusters and transforms each cluster into one predictor. A small subset of the transformed predictors informative for prediction is then selected for each locus to construct a regression model ("Methods"). Clustering reduces the predictor dimension, mitigates co-linearity, and makes the predictors less sensitive to measurement noise[13]. Compared to using individual genes as predictors, clustering improved the prediction accuracy ("Methods", Supplementary Fig. 4). Conceptually, constructing regression models using selected gene clusters is a group variable selection approach applied in a prediction setting[14]. However, unlike $BIR(\overline{X}, Y)$, conventional group variable selection methods are primarily developed for modeling a univariate response using high-dimensional predictors[15–17]. They pay less attention to important issues such as computational efficiency and robustness to noisy predictors for handling complex big data where both predictors and responses are high-dimensional. We compared $BIR(\overline{X}, Y)$ with the popular group variable selection methods

group lasso[16] and composite minimax concave penalty regression[17] (composite MCP). $\mathrm{BIR}(\overline{X}, Y)$ was more accurate and computationally efficient. We also compared $\mathrm{BIR}(\overline{X}, Y)$ with fused lasso[15] based on 1% of the genome. Both methods yielded similar prediction accuracy, but fused lasso was >$10^5$ times slower than $\mathrm{BIR}(\overline{X}, Y)$ (Supplementary Fig. 5; Supplementary Note 1).

The pathway-level model, denoted by $\mathrm{BIR}(\overline{X}, \overline{Y})$, not only clusters correlated predictors but also groups correlated responses (i.e., co-activated DHSs) into clusters. It predicts the mean DH level of each cluster. Each DHS cluster can be viewed as a "pathway" consisting of co-activated regulatory elements[18, 19], and the mean DH level of all loci in a pathway can be viewed as the pathway's activity. Because using the cluster mean reduces the variance of measurement noise, pathway activities are less noisy

than locus-level DH measurements. Thus, $\mathrm{BIR}(\overline{X}, \overline{Y})$ was able to predict pathway activities much more accurately than predicting DH levels of individual loci using the locus-level model $\mathrm{BIR}(\overline{X}, Y)$ (Supplementary Fig. 7).

The final prediction by BIRD for each genomic locus is a weighted average of the locus-level and pathway-level predictions. The intuition is that the pathway activity predicted by $\mathrm{BIR}(\overline{X}, \overline{Y})$ can also serve as a prediction of the DH level for each individual locus within the pathway. Such a prediction may be biased but less noisy than the locus-level prediction by $\mathrm{BIR}(\overline{X}, Y)$. Thus, integrating the locus-level prediction from $\mathrm{BIR}(\overline{X}, Y)$ and the pathway-level prediction from $\mathrm{BIR}(\overline{X}, \overline{Y})$ through model aggregation may result in a better tradeoff between the prediction bias and variance. In fact, the aggregated model robustly improved the overall locus-level prediction accuracy (Supplementary Fig. 4,

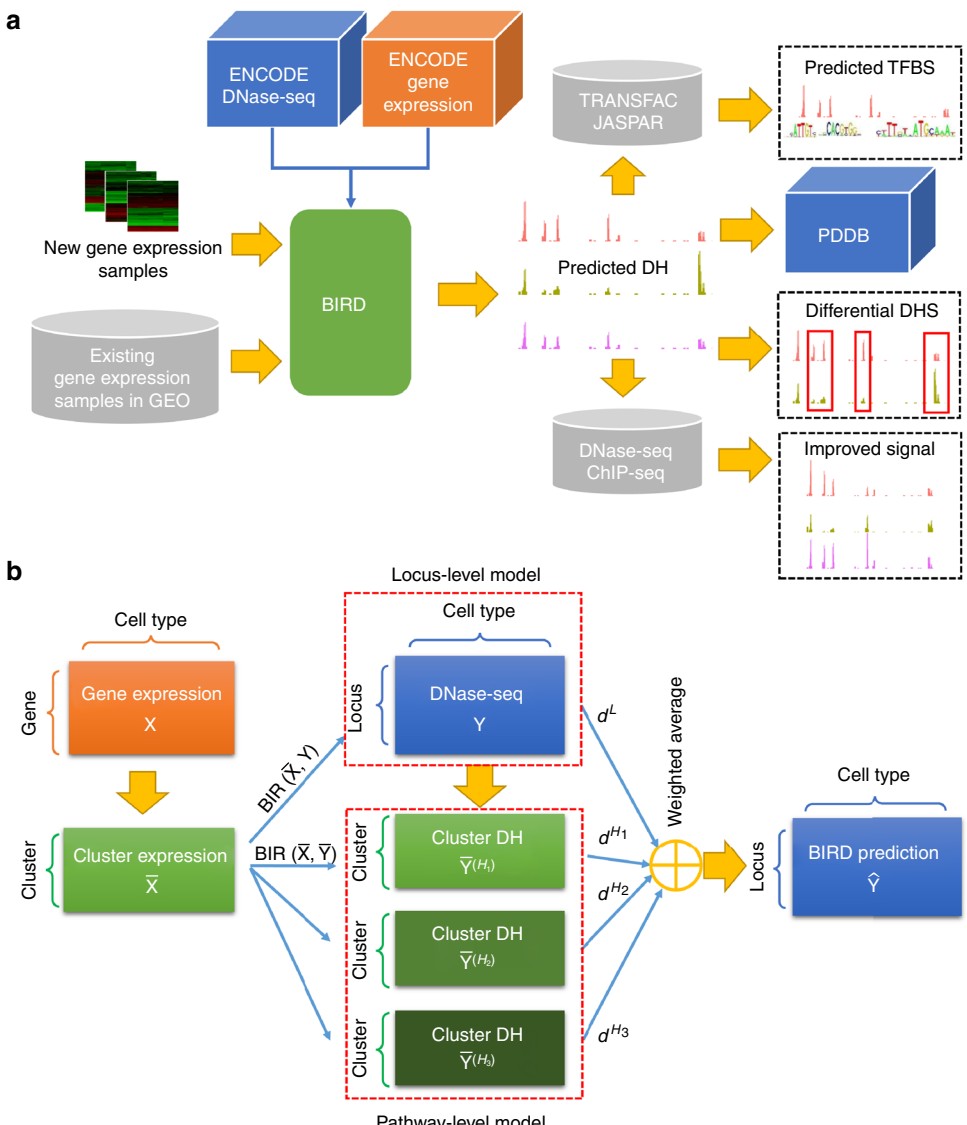

**Fig. 1** Concepts of BIRD. **a** Outline of the study. ENCODE DNase-seq and exon array data are used to train BIRD. Users can apply BIRD to new or existing gene expression samples to predict DH. The predicted DH can be used to predict TFBSs and differential DHSs, convert expression samples in GEO into a regulome database (PDDB), and improve DNase-seq and ChIP-seq data analyses. **b** Overview of BIRD. Instead of using individual genes as predictors, BIRD groups co-expressed genes into clusters (i.e., gene-cluster) and uses the clusters' mean expression levels as predictors. BIRD aggregates two types of models. The locus-level model $\mathrm{BIR}(\overline{X}, Y)$ predicts the DH level at each genomic locus. The pathway-level model $\mathrm{BIR}(\overline{X}, \overline{Y})$ further groups correlated loci (i.e., loci with co-varying DH) into different levels of clusters (i.e., DHS pathways) and predicts the DH level for each pathway. Finally, BIRD predicts DH at each locus by combining the locus-level and pathway-level predictions via model averaging

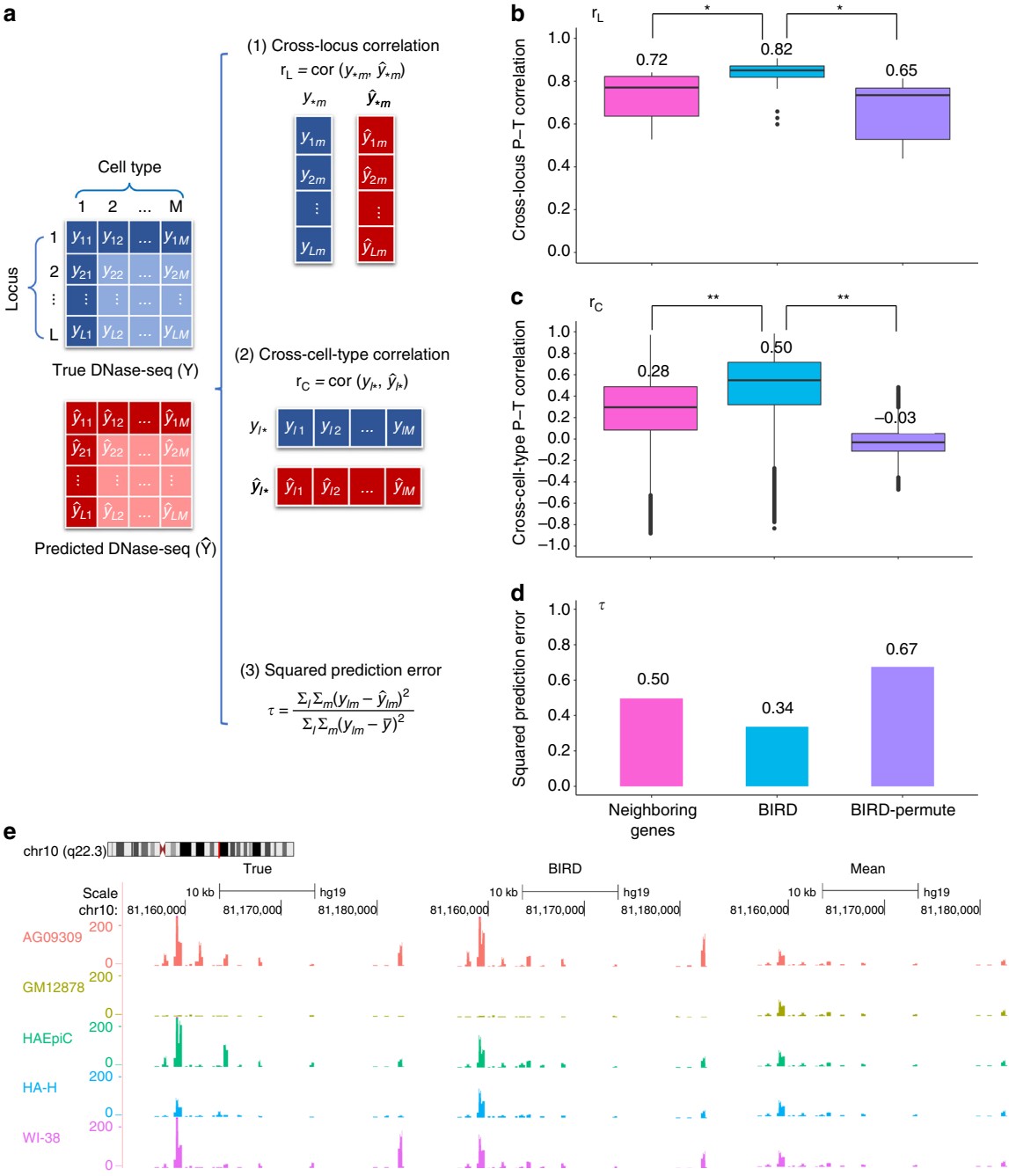

**Fig. 2** Evaluation of BIRD. **a** Statistics used to evaluate prediction accuracy. **b** Cross-locus P–T correlation ($r_L$) for different methods. For each method, the distribution and mean of $r_L$ from the 17 test cell types are shown. *Two-sided Wilcoxon signed-rank test $p$-values $< 10^{-4}$ for comparing two methods ($n = 17$ for each test). **c** Cross-cell-type P–T correlation ($r_C$) for different methods. For each method, the distribution and mean of $r_C$ from the 912,886 genomic loci are shown. **Two-sided Wilcoxon signed-rank test $p$-values $< 10^{-15}$ for comparing two methods ($n = 912,886$ for each test). **d** Squared prediction error ($\tau$) for different methods. Wilcoxon signed-rank test was not conducted here since there is only one $\tau$ for each method. **e** An example of true and predicted DNase-seq signals for five different cell types. "True": true DNase-seq bin read count; "BIRD": DH signal predicted by BIRD; "Mean": the average DH signal of training cell types. For "BIRD" and "Mean", signals are transformed back from the log-scale to the original scale. (**b**, **c**) The box in each boxplot denotes the 1st (Q1) to 3rd (Q3) quartile of the data, the line in the middle of the box is the median, and the lower and upper whiskers represent the 1.5 interquartile range from Q1 and Q3, respectively

"Methods"). On the basis of these results, we used the aggregated model, termed BIRD, in all subsequent analyses to predict DH levels for each individual locus. A systematic benchmark analysis shows that BIRD not only produces the best prediction performance compared to other methods, but also offers computational efficiency suitable for big data regression (Supplementary Note 1; Supplementary Figs. 4–6).

**Predicting DH based on gene expression**. After building BIRD prediction models using the 40 training cell types, we evaluated their prediction performance in the 17 test cell types. Below is a summary of the major findings.

Gene expression provides valuable information for predicting DH. Compared to random prediction models, BIRD significantly increased the P–T correlation ($r_L$ and $r_C$) and substantially

decreased the squared prediction error ($\tau$) (Fig. 2b–d: BIRD vs. BIRD-Permute; "Methods"). Here random prediction models were constructed by applying BIRD to the training data after permuting the link between DNase-seq and gene expression samples.

Prediction based on the whole transcriptome substantially improves prediction based on a locus' neighboring genes. We tested the neighboring gene approach by gradually increasing the number of neighboring genes ("Methods", Supplementary Fig. 2). Compared to the best performance of the neighboring gene approach, BIRD produced substantially higher prediction accuracy (Fig. 2b–d), indicating that not all information useful for prediction is contained in neighboring genes. This is reasonable as many regulatory elements are known to control genes over a long genomic distance, sometimes across many other genes. Also, DH of a locus may be correlated in trans with expression of TFs that bind to the locus, genes that co-express with these TFs, and genes that co-express with the target gene controlled in cis by the locus. Moreover, since cell-type-specific transcription of a gene may be controlled by multiple cis-regulatory elements, a gene's expression may not always correlate well with the DH level of each individual regulatory element in its neighborhood.

DH variation across different genomic loci within a cell type can be accurately predicted. In the 17 test cell types, the mean cross-locus P–T correlation $r_L$ for BIRD was 0.82 (Fig. 2b). Notably, random prediction models were also able to produce large $r_L$ (Fig. 2b, mean = 0.65). This is because different loci have different DH propensity, consistent with observations in a previous study[12]. For instance, some loci tend to show higher DH signal than other loci in most cell types (Supplementary Fig. 8). Thus, using the average DH profile of all training cell types can predict the cross-locus DH variation in a new cell type with good accuracy[12], even though such predictions are cell-type-independent and remain the same for all new cell types. Our random prediction models were generated by permutations that did not perturb the locus-specific DH propensity. Therefore, their $r_L$ was large. Since BIRD uses cell-type-dependent information carried by transcriptome, its predictions are more accurate (Fig. 2b).

DH variation across cell types can be predicted, although it is more challenging than predicting cross-locus variation. Figure 2e shows an example demonstrating that the true cross-cell-type DH variation measured by DNase-seq can be captured by BIRD predictions, but not by the mean DH profile of all training cell types. Comparing the cross-locus P–T correlation ($r_L$) in Fig. 2b with the cross-cell-type P–T correlation ($r_C$) in Fig. 2c, $r_L$ on average was much larger than $r_C$ (0.82 vs. 0.50 for BIRD). Unlike $r_L$, the distribution of $r_C$ for random prediction models was centered around zero (Fig. 2c, mean = −0.03) because the cross-cell-type prediction accuracy was evaluated within each locus and hence not affected by locus effects. Compared to random prediction models, BIRD substantially increased $r_C$ (Fig. 2c).

Cross-cell-type prediction accuracy varies greatly among different loci. For 6% of loci, BIRD predictions had $r_C < 0$ (i.e., prediction did not help). On the other hand, 56 and 20% of loci had $r_C > 0.5$ and >0.75, respectively, indicating that DH could be predicted with moderate to high accuracy for a substantial fraction of loci. By examining the true DH levels measured by DNase-seq in the training and test cell types, we found that multiple factors may influence cross-cell-type prediction accuracy of a locus. First, a subset of loci was not active in any test cell type. For these loci, the true DH levels are essentially noise. The cross-cell-type correlation between the predicted DH levels and random noise is expected to be low. Therefore, these noisy loci are not informative for evaluating the performance of predicting biological variation across cell types. After excluding these noisy loci, we found that loci with low signal range (characterized by the difference between the maximal and

minimal DH values), low signal variability (characterized by coefficient of variation (CV)), or high cell-type-specificity (characterized by the number of cell types in which the locus is active or inactive) tend to have lower $r_C$ (Fig. 3a; Supplementary Fig. 9; Supplementary Methods). Altogether, these factors and noisy loci explained the majority (85%) of loci with low $r_C$ (i.e., $r_C$ < 0.25) (Fig. 3b–d). In real applications, BIRD is most useful for making predictions in new cell types for which DNase-seq data are not available. Therefore, we repeated this analysis by using BIRD-predicted DH levels (instead of true DH levels) in the test cell types and the true DH levels in training cell types for locus stratification. The analysis produced similar results (Supplementary Fig. 10). In practice, one may use the factors discussed above to screen for loci whose cross-cell-type prediction is likely to be accurate. For instance, if one filters out the noisy loci and loci with low max–min spread, low CV or high cell-type-specificity based on true DH values in the training data and predicted DH values in the test data, the mean $r_C$ would become 0.6 (compared to the mean of 0.5 for all loci, and 0.43 for filtered loci), and 74 and 30% of loci would have $r_C > 0.5$ and >0.75, respectively (Fig. 3e; Supplementary Methods). We also investigated the relationship between $r_C$ and the mean DH level of a locus but did not find strong correlation between them (Supplementary Methods; Supplementary Fig. 9e, f).

Cross-cell-type DH variation of regulatory element pathways can be predicted with substantially higher accuracy than that of individual loci. As a building block of BIRD, the pathway-level model $\mathrm{BIR}\left(\overline{X}, \overline{Y}\right)$ groups correlated DHSs into clusters based on the training data. Treating each cluster as a pathway, $\mathrm{BIR}\left(\overline{X}, \overline{Y}\right)$ predicts the activity of each pathway (i.e., the mean DH level of all DHSs in each cluster) ("Methods"). When DHSs were grouped into 1000 clusters, the cross-cell-type P–T correlation $r_C$ for the pathway-level prediction by $\mathrm{BIR}\left(\overline{X}, \overline{Y}\right)$ was substantially higher than $r_C$ for the locus-level prediction by BIRD (Fig. 3f, mean $r_C$ for $\mathrm{BIR}\left(\overline{X}, \overline{Y}^{(1000)}\right)$ vs. BIRD = 0.71 vs. 0.50). For $\mathrm{BIR}\left(\overline{X}, \overline{Y}^{(1000)}\right)$, 84 and 55% clusters had $r_C > 0.5$ and >0.75 respectively. Similar results were obtained when DHSs were grouped into 2000 or 5000 clusters (Fig. 3f, $\mathrm{BIR}\left(\overline{X}, \overline{Y}^{(2000)}\right)$ and $\mathrm{BIR}\left(\overline{X}, \overline{Y}^{(5000)}\right)$). Thus, similar to gene set analyses[20], the overall cross-cell-type activity of a DHS pathway can be more reliably studied using prediction than that of individual loci.

Prediction of differential DH is feasible. As one potential application of BIRD is to predict differential DH between two sample types, we further evaluated BIRD by conducting pairwise comparisons of the 17 test cell types. Cell type pairs were stratified into four equal-sized groups based on the similarity level of the global DH profiles of two compared cell types. For each pair of cell types, the correlation between the predicted and true differential DH was computed across all loci and across differential loci, respectively ("Methods"). For cell type pairs in the highest similarity quartile, the mean correlation was 0.42 for all loci and 0.55 for differential loci (Fig. 3g). For cell type pairs in the other similarity quartiles, the mean correlation was higher: 0.60–0.66 for all loci and 0.69–0.75 for differential loci (Fig. 3g). Again, compared to the prediction for individual loci, differential DH prediction at pathway-level was more accurate (Fig. 3g, mean correlation = 0.60–0.77; Supplementary Fig. 11).

The conclusions above do not depend on how the 57 cell types are partitioned into the training and testing data. We repeated the same analyses on four other random partitions ("Methods", Supplementary Data 1), and similar results were obtained. For

instance, Supplementary Fig. 12 shows that $r_L$, $r_C$, and $\tau$ for BIRD from different partitions were similar. The conclusions remained qualitatively the same when Spearman's rank correlation was used instead of Pearson's correlation ("Methods"; Supplementary Figs. 13 and 14).

**Comparisons of BIRD and ChromImpute.** ChromImpute is a method for imputing one functional genomic data type using multiple other data types[12]. We compared DH predictions by BIRD using only gene expression data with DH predictions by ChromImpute using multiple functional genomic data types (Supplementary Methods). Among 10 tested cell types, BIRD and ChromImpute showed comparable cross-locus and cross-cell-type prediction performance. Neither method consistently outperformed the other (Fig. 3h, i; Supplementary Fig. 15a–f). We further applied both methods to predict differential DH between each pair of the 10 test cell types. BIRD outperformed

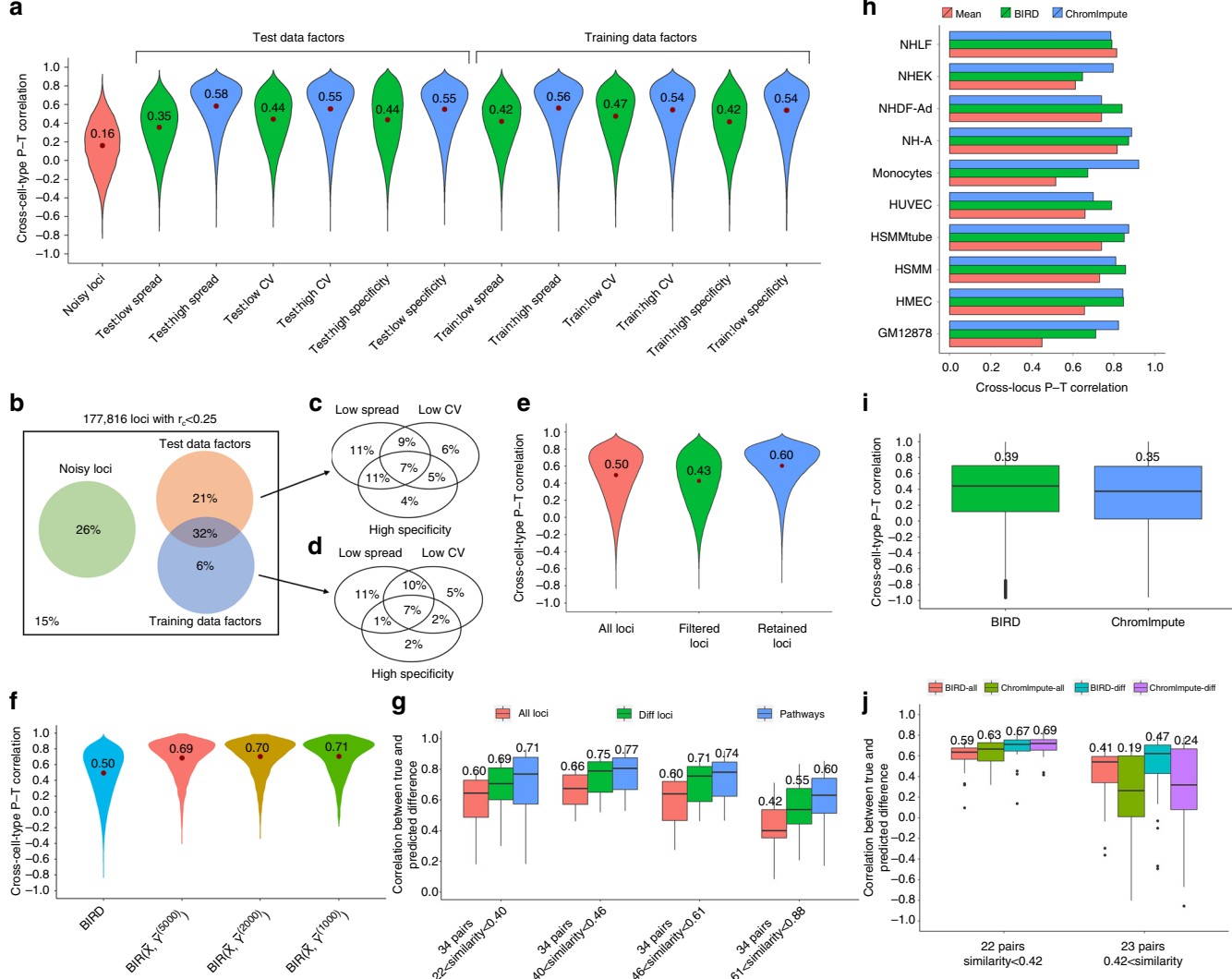

**Fig. 3** Cross-cell-type prediction performance and a comparison with ChromImpute. **a** Distribution and mean of $r_C$ for different loci classes. DHSs were categorized into noisy loci and non-noisy loci based on test data (Supplementary Methods). The non-noisy loci were divided into two groups (low or high) based on different factors: max–min spread, coefficient of variation (CV), cell-type-specificity. The grouping was done separately using the true DH values from the test data and from the training data. **b–d** Percentage of loci with low cross-cell-type prediction accuracy ($r_C < 0.25$) explained by noisy loci, low max–min spread, low CV, high cell-type-specificity in test cell types (details shown in **c**) and training cell types (details shown in **d**). **e** Distribution of $r_C$ for all loci, loci filtered out by factors in (**b**) (i.e., noisy loci, low max–min spread, low CV, or high cell-type-specificity), and the remaining loci. **f** Comparison between locus-level and pathway-level predictions in terms of cross-cell-type prediction accuracy. For each method, $r_C$ distribution of all genomic loci or pathways (5000, 2000, and 1000) are shown. **g** Accuracy for predicting differential DH between two cell types. The 136 pairs of test cell types were stratified based on the quartiles of the similarity between the two compared cell types. For each stratum, the distribution and mean of the prediction–truth Pearson's correlation across all loci, differential loci, and DHS pathways (1000) are shown. **h–j** Comparison between BIRD and ChromImpute using 10 test cell types. **h** Cross-locus P–T correlation $r_L$. As a baseline, predictions based on the mean DH profile of training cell types are shown. **i** Cross-cell-type P–T correlation $r_C$. **j** Accuracy for predicting differential DH between two cell types. Test cell type pairs were divided into two groups based on the median of similarity between the two compared cell types. For each group, the distribution and mean of the prediction–truth correlation across all loci and differential loci are shown. (**g**, **i**, **j**) The boxplots show median (central line), interquartile range (IQR, the 1st (Q1) to 3rd (Q3) quartiles, box), and 1.5 × IQR from the Q1 and Q3 (lower and upper whiskers), respectively

ChromImpute substantially for comparing cell types with high similarity (i.e., cell-type pairs whose similarity level was above the median), and ChromImpute performed slightly better than BIRD for comparing cell types with lower similarity levels (Fig. 3j; Supplementary Fig. 15g–i). Note that ChromImpute used ChIP-seq data for multiple histone modifications as predictors (these are the best predictors selected by ChromImpute for imputing DH[12]), which are non-trivial to generate. By contrast, BIRD was based on gene expression data, which are easier to generate and widely available.

**Predictors selected by BIRD.** Predictors of each DHS in BIRD consist of predictors from the locus-level model and the pathway-level models. Analyses of these predictors show that only a small proportion of DHSs had their closest genes or target genes contained in the predictors, although this proportion was significantly larger than random expectation (permutation test p-values < 0.001) (Supplementary Note 2; Supplementary Fig. 16a–c). For DHSs with phylogenetically conserved DNA sequences, we did not found their predictors to be more conserved (Supplementary Note 3; Supplementary Fig. 17).

We further analyzed predictors selected by the pathway-level models (i.e., $BIR(\overline{X}, \overline{Y})$) for each DHS pathway. We found that TFs that potentially regulate the pathways were enriched in pathways' predictors (Supplementary Note 2; Supplementary Fig. 16d–j; Supplementary Data 2). The complete catalog of BIRD-selected predictor genes for each DHS and pathway, and the DNA motifs and GO terms enriched in the predictors are provided at https://zhiji.shinyapps.io/CABS/ as an online resource.

**Predicting TFBSs.** One demonstrated application of DNase-seq is to predict TFBSs by coupling DH with DNA motif information[21, 22]. We asked whether the BIRD-predicted DH can be used for this task when experimental DNase-seq data are not available. Using BIRD models based on the 40 training cell types, we predicted TFBSs for nine TFs in GM12878 cell line, which was not in the training data. For evaluation, reproducible binding peaks from the corresponding ENCODE TF ChIP-seq data in the same cell line were downloaded. As it is unrealistic to expect a prediction algorithm to identify which TFs bind to a genomic site without motif information, we made our predictions for each TF at its motif sites (Supplementary Data 3). Correspondingly, reproducible ChIP-seq peaks that contained the TF's motif were used as gold standard for evaluation. We used motif-containing peaks as gold standard also because ChIP-seq peaks without motifs may correspond to indirect TF-DNA association. As controls, we predicted TFBSs using motif sites alone ("Motif"; negative control), the mean DH profile of all training cell types at motif sites ("Mean"), and true DNase-seq data at motif sites ("True"; positive control). Different methods were compared based on the sensitivity at different false discovery rate (FDR) levels (Fig. 4a; Supplementary Fig. 18), the receiver operating characteristics (ROC, i.e., true-positive rate vs. false-positive rate) (Fig. 4b; Supplementary Fig. 19), and the number of predicted binding sites at different FDR levels (Fig. 4c; Supplementary Fig. 20). Figure 4d–f shows the area under the sensitivity–FDR curve (AUSFC), the area under the ROC curve (AUROC), and the number of predicted TFBSs at the 50% FDR level. Here the 50% FDR was merely used to provide a snapshot to compare different methods. One can reduce FDR by reducing the number of predictions and sensitivity as shown in Fig. 4a, c and Supplementary Figs. 18 and 20. As expected, TFBS prediction based on true DNase-seq data was more accurate than BIRD (Fig. 4d–f). However, BIRD substantially improved TFBS prediction based on the motif only or mean DH methods.

Taking ELF1 as an example, at 10, 25, and 50% FDR level, BIRD (UW) predictions gave a sensitivity of 0.43, 0.64, and 0.88, respectively, as compared to 0.27, 0.64, and 0.94 by the true DNase-seq approach, 0, 0.02, and 0.11 by the motif only approach, and 0.09, 0.36, and 0.62 by the mean DH approach (Fig. 4a). At these FDR levels, BIRD generated 5000, 9000, and 19,000 predicted ELF1 binding sites, which were close to the true DNase-seq approach (3000, 9000, and 20,000) and substantially more than the number of the predicted sites by the motif only (0, 250, and 2000) and mean DH (1000, 5000, and 13,000) methods (Fig. 4c). The area under the sensitivity–FDR curve and ROC curve (AUSFC and AUROC, Fig. 4d, e) for BIRD (AUSFC = 0.61, AUROC = 0.93) were also close to the true DNase-seq approach (AUSFC = 0.62, AUROC = 0.95) and substantially better than the motif only (AUSFC = 0.16, AUROC = 0.65) or the mean DH (AUSFC = 0.43, AUROC = 0.82) approach. Figure 4g and h shows predictions in two genomic regions, which illustrate how BIRD better predicts TFBSs than the mean DH and motif only methods.

In our analyses, BIRD predictions were made using exon array data generated by three different laboratories. The lab difference turned out to be smaller than the differences between prediction methods (Fig. 4a–c; Supplementary Figs. 18–20). We also compared our methods with two state-of-the-art TFBS prediction methods PIQ[22] and CENTIPEDE[21]. Both PIQ and CENTIPEDE use true DNase-seq data and motif information to make predictions. PIQ showed comparable performance with our true DNase-seq method, whereas CENTIPEDE performed worse than BIRD (Supplementary Methods; Supplementary Figs. 21–24). Thus, replacing our true DNase-seq method by PIQ or CENTIPEDE as positive controls did not change the main conclusion regarding the usefulness of BIRD.

We conducted similar analyses to three TFs in another cell line, K562, and obtained similar results (Supplementary Figs. 25–28). In addition, we applied BIRD to a non-ENCODE cell line, P493-6 B cell lymphoma, to predict MYC binding sites. For this dataset, no corresponding DNase-seq data were available. Thus, PIQ, CENTIPEDE, and prediction based on true DH were not applicable. We compared BIRD with the mean DH and motif only methods. BIRD again outperformed the other two methods (Supplementary Methods; Supplementary Fig. 29).

Our analyses indicate that when experimental regulome data are not available, BIRD-predicted DH can be used to predict TFBSs. One limitation of this approach is that it requires accurate motif information and the prediction is contingent on the presence of TF binding motifs. This limitation, however, is not unique to BIRD. It is common to methods such as PIQ and CENTIPEDE that use chromatin accessibility to predict TFBSs. Despite this limitation, BIRD can generate a substantial amount of information (e.g., thousands of predicted TFBSs) one would not have without BIRD. We note that the prediction performance presented above does not represent the upper limit one can achieve using BIRD. It is possible to further improve TFBS prediction by incorporating other information (e.g., by using a more sophisticated motif model that accounts for intra-motif correlation) (Supplementary Note 4; Supplementary Fig. 30). However, systematically exploring the optimal use of non-DH predictors is beyond the scope of this study.

**Regulome prediction using public expression samples in GEO.** The vast amounts of gene expression data from diverse biological contexts in GEO represent a resource that no single laboratory can generate. We next asked whether one can use BIRD to turn this resource on gene expression into a resource on regulome studies. As a proof of concept, we collected 2000 human exon

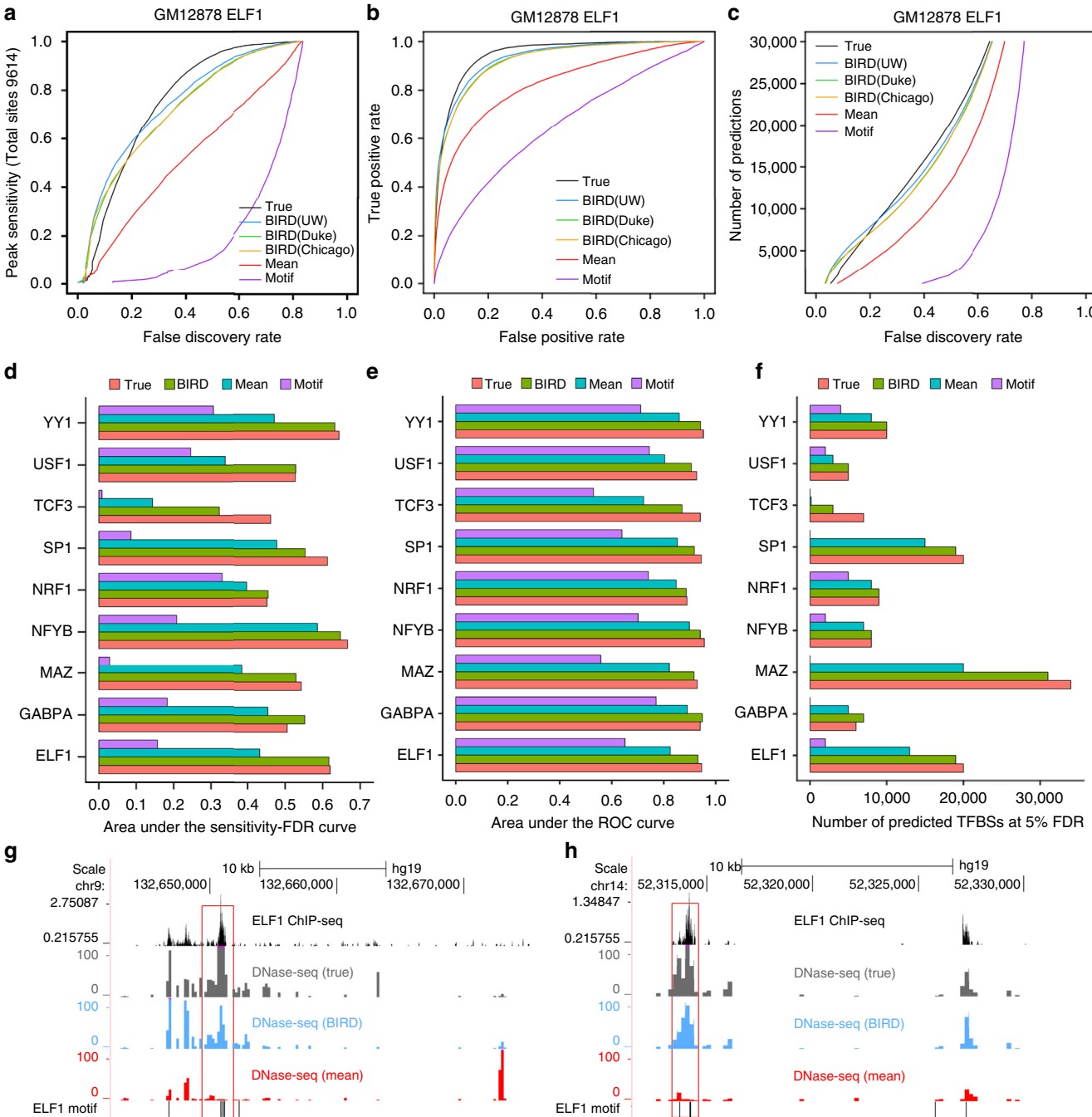

**Fig. 4** Predicting transcription factor binding sites. **a** Sensitivity–FDR curve for predicting ELF1 binding sites in GM12878 using four different methods: true DNase-seq data ("True"), BIRD, mean DH profile of training cell types ("Mean"), and motif mapping score ("Motif"). For BIRD, "BIRD(UW)","BIRD(Duke)", and "BIRD(Chicago)" denote predictions made using exon arrays generated by three different labs. For each method, the sensitivity–FDR curve shows the percentage of gold standard TFBSs that were discovered by the predicted binding sites at different FDR levels. The total number of gold standard TFBSs was shown on the *y* axis in the brackets. **b** ROC curve for predicting ELF1 binding sites in GM12878 using different methods. **c** The number of DHSs predicted to be ELF1 binding sites in GM12878 by different methods at different FDR levels. **d** Area under the sensitivity–FDR curve for predicting TFBSs of nine TFs in GM12878 using "True", "BIRD" (based on BIRD(UW)), "Mean", and "Motif". **e** Area under the ROC curve for predicting TFBSs of nine TFs in GM12878 using different methods. **f** Number of predicted TFBSs at 50% FDR for predicting TFBSs of nine TFs in GM12878 using different methods. **g**, **h** Two examples showing the true ELF1 ChIP-seq signal (read count, black) in GM12878, the true DNase-seq signal (gray), the BIRD-predicted DH signal (blue), and the "Mean" DH signal in training cell types (red). Locations of ELF1 motif sites are shown on the bottom. BIRD more accurately captured the true signal than "Mean" (highlighted with red boxes). It also predicts TFBSs better than the motif only approach as many motif sites are not bound and do not have DH signal at all

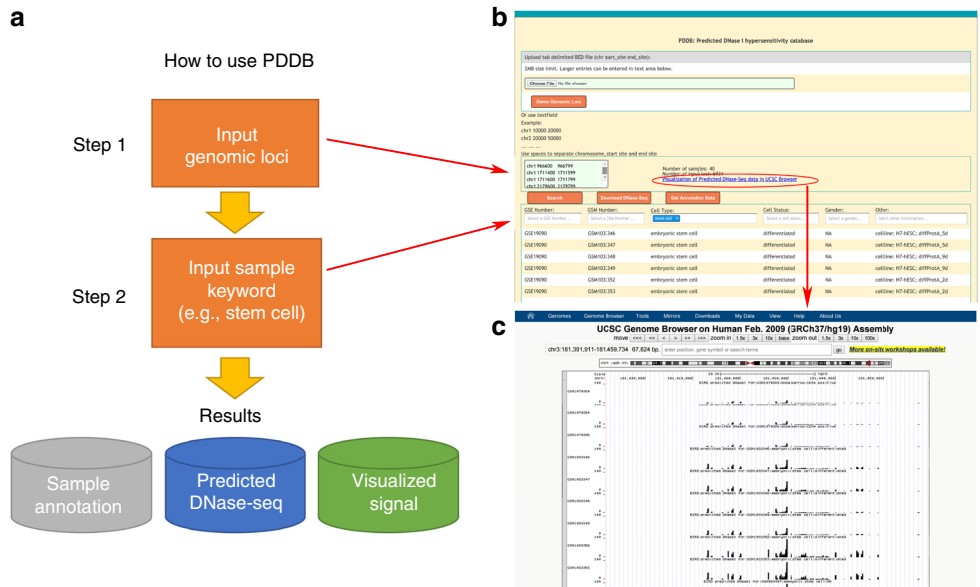

**Fig. 5** The predicted DNase I hypersensitivity database (PDDB). **a** Flowchart illustrating how to use PDDB. Step 1: provide a list of genomic loci of interest. Step 2: provide keywords in one or multiple annotation fields (e.g., type "stem cell" in the "Cell Type" column) to search for samples of interest. PDDB will return predicted DH for the queried loci and samples along with sample annotation and data for visualization. **b** Web interface of PDDB. Users can download the predicted DH data by clicking the "Download DNase-seq" button. The sample annotation data can be downloaded by clicking the "Get Annotation Data" button. **c** By clicking the "Visualization of Predicted DNase-Seq data in UCSC Browser" link in the PDDB web interface (red circle in **b**), one can display the predicted DH signal in the UCSC genome browser

array samples from GEO and applied BIRD trained using all 57 ENCODE cell types for 1,108,603 loci to these samples to predict regulome. These predictions are made available as a web resource PDDB (Predicted DNase I hypersensitivity database). A user interface is provided for data query, display and download (Fig. 5, "Methods").

Researchers can use PDDB to explore regulatory element activities in biological contexts for which they do not have available regulome data. As a feasibility test, we first queried predicted DH for three genes FBL, LIN28A and BLMH in P493-6 B cell lymphoma (for which no public DNase-seq data are available) and H9 human embryonic stem cells. Promoters of these genes are known to be bound by MYC in a cell type-dependent fashion[23]. FBL is bound in both P493-6 and H9, LIN28A is bound in H9 but not in P493-6, and BLMH is bound in P493-6 but not in H9[23–25]. PDDB successfully predicted these known cell-type-dependent binding patterns (Fig. 6a–c; Supplementary Fig. 31).

Next, we obtained SOX2 binding sites in human embryonic stem cells from a published ChIP-seq study[26] (Supplementary Methods; Supplementary Data 4). Figure 6d shows the predicted DH at these sites across the 2000 GEO samples. The samples were ordered based on the overall DH enrichment level at all SOX2 binding sites relative to random genomic sites (Supplementary Methods; Fig. 6e). Samples with strong predicted DH at SOX2 binding sites include stem cells (green bar in Fig. 6d) and brain (brown bar), consistent with known roles of SOX2 in these sample types[27–30]. Interestingly, PDDB contained differentiating H7 embryonic stem cells collected at day 2, 5, and 9 after initiation of differentiation. Our 57 training cell types contained undifferentiated H7 cells and H7 cells at differentiating day 14. Altogether, these samples formed a time course. Examination of the predicted DH for day 2, 5, and 9 along with the true DH for day 0 and 14 shows that the predicted DH at SOX2 binding sites decreased as the differentiation progressed (Fig. 6f, g), consistent with the known role of SOX2 in maintaining the undifferentiated

status of stem cells[27, 28]. Thus, the differential activities of SOX2 binding were correctly predicted in PDDB.

The above examples show that expression samples in GEO can be used to meaningfully predict DH. With ChIP-seq data for a TF from one biological context, one may use PDDB to systematically explore in what other biological contexts each binding site might be active, and group TFBSs into functionally related subclasses accordingly. For instance, we obtained MEF2A ChIP-seq binding sites in GM12878 lymphoblastoid cells from ENCODE. MEF2A is a TF involved in muscle development[31] and neuronal differentiation[32]. Using PDDB (Supplementary Methods; Fig. 6h, i; Supplementary Fig. 32; Supplementary Data 5–7), a group of MEF2A binding sites associated with genes involved in cell motion, cell migration and regulation of metabolic processes were found to be more active in muscle-related samples (including coronary artery smooth muscle and cardiac precursor cell which are not covered by ENCODE) than in lymphoblastoid (Fig. 6h, i). Another group of sites associated with neuron differentiation and neurogenesis genes were found to be more active in neuron and brain-related samples (including non-ENCODE sample types such as entorhinal cortex and motor neuron) (Fig. 6h, i). This demonstrates how PDDB can provide a detailed view of TFBSs not offered by the original experiment in GM12878, and how PDDB can be used to investigate many biological contexts not covered by ENCODE.

**Predicting differential signals in a differentiation system**. To test the application of BIRD to studying differential DH, we analyzed a differentiation system in which human induced pluripotent stem cells (iPSCs) were treated to differentiate into dopaminergic neurons[33] (Supplementary Methods). For this system, exon array samples were established from iPSCs and iPSC-derived neurons, but no corresponding experimental regulome data were available. This represents a typical scenario for which BIRD is useful. Using BIRD, we predicted differential DH and

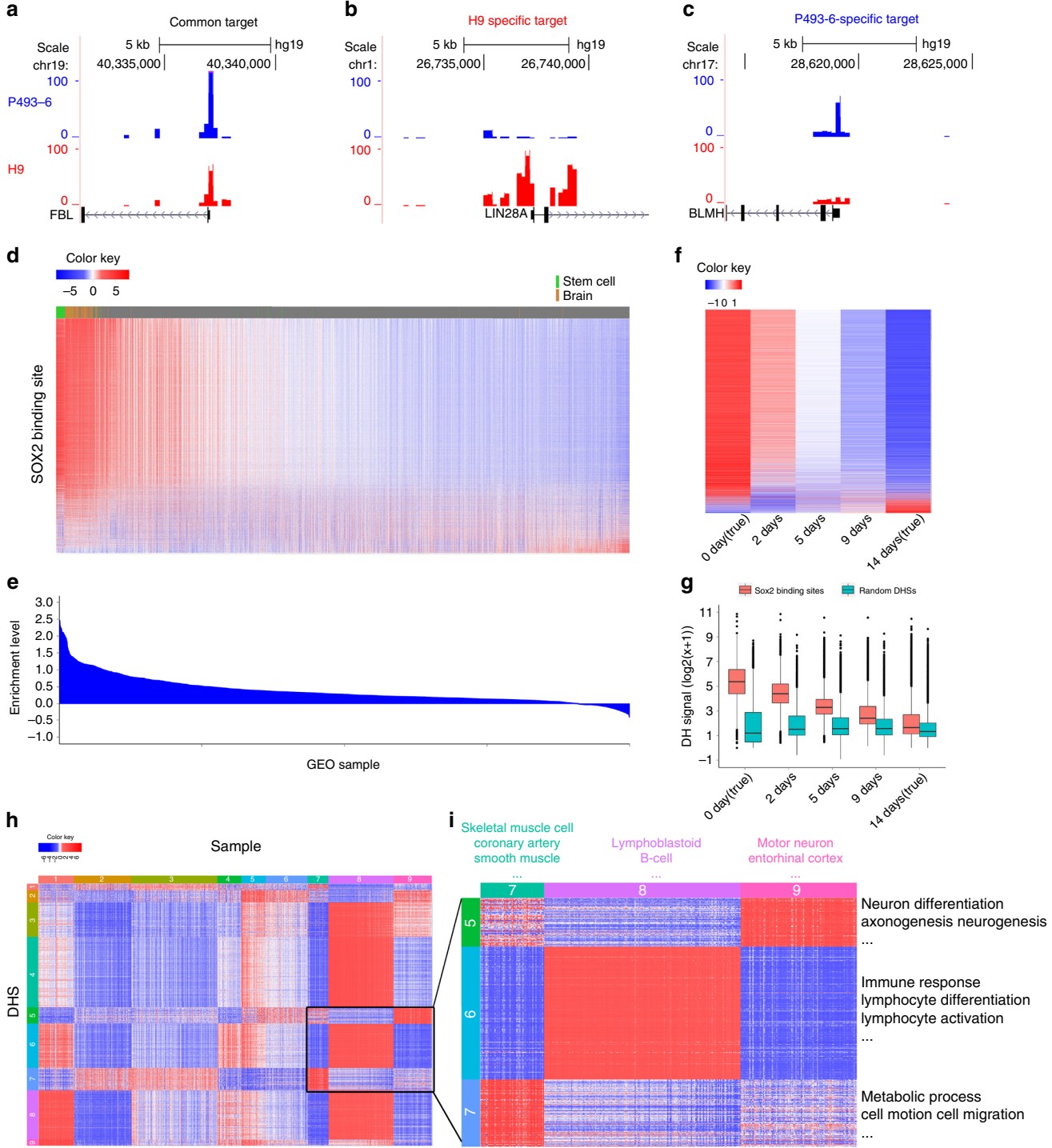

**Fig. 6** Predicting regulome using PDDB. **a–c** Predicted DH in promoter regions of *FBL* (**a**), *LIN28A* (**b**), and *BLMH* (**c**) in P493-6 B cell lymphoma and H9 embryonic stem cells. For H9, PDDB contains multiple replicate samples, which produced similar results. One replicate is shown here and the other replicates are shown in Supplementary Fig. 31. **d** Predicted DH at 6931 SOX2 binding sites in 2000 PDDB samples. Each column is a sample, and each row is a binding site. Values within each row are standardized to have zero mean and unit SD before visualization. **e** Relative DH enrichment level when comparing SOX2 binding sites with random sites (Supplementary Methods). **f** Predicted DH at SOX2 binding sites in H7 stem cells after 2, 5, and 9 days of differentiation. True DH from undifferentiated H7 cells and cells at differentiating day 14 in the training data are also shown. Rows are SOX2 sites and columns are time points. Values within each row are standardized before visualization. **g** Predicted DH at SOX2 binding sites are compared with predicted DH at 10,000 random DHSs. At each time point, DH values from all sites are displayed using a boxplot, which shows the median (central line), interquartile range (IQR, the 1st (Q1) to 3rd (Q3) quartiles, box), and 1.5 × IQR from the Q1 and Q3 (lower and upper whiskers), respectively. **h** Predicted DH at 2011 MEF2A binding sites in 1061 *MEF2A*-expressing PDDB samples (Supplementary Methods). Each column is a sample. Each row is a MEF2A binding site. Values within each row are standardized before visualization. Samples and DHSs were clustered. **i** The highlighted region in **h** that shows DHS-clusters with increased DH in muscle, lymphoblastoid, and brain-related samples, respectively

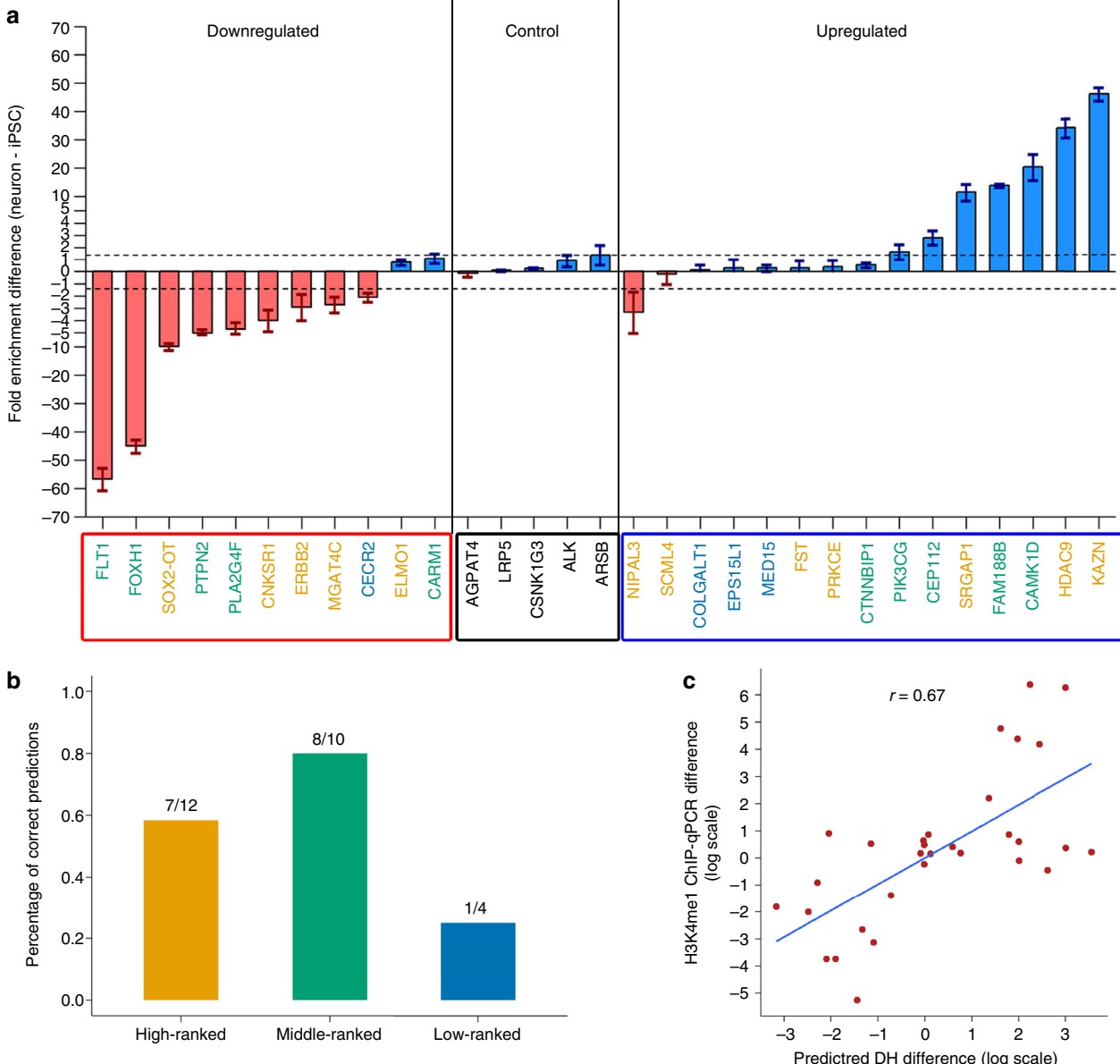

**Fig. 7** Predicting differential activities of regulatory elements during neuronal differentiation. **a** Difference in H3K4me1 measured by ChIP-qPCR between iPSCs and iPSC-derived neurons for 26 DHSs at different predicted fold change levels (i.e., absolute value of log2 fold change $|\delta|>2$, $1<|\delta|\leq 2$, and $0.1 <|\delta|\leq 1$, indicated by yellow, green, and blue, respectively, in the gene labels) and five non-differential DHSs ("control") predicted by BIRD. The 26 DHSs were further divided into up regulated and downregulated DHSs after differentiation based on the predicted DH. Dashed lines represent ± maximum IChIP-qPCR measured H3K4me1 difference in five control DHSsl. Data are presented as mean ± s.e.m. (iPSCs and iPSC-derived neurons each had $n=3$ technical replicates). **b** Validation rate of differential DHSs: 7/12 high-ranked ($|\delta|>2$) DHSs, 8/10 middle-ranked ($1<|\delta|\leq 2$) DHSs, and 1/4 low-ranked ($0.1<|\delta|\leq 1$) DHSs were validated by ChIP-qPCR. **c** Correlation between the predicted differential DH (i.e., difference in the predicted DH signals (at log2 scale) between iPSCs and iPSC-derived neurons) and the ChIP-qPCR measured H3K4me1 difference (log2 fold change) across the 31 DHSs tested by ChIP-qPCR. Each dot is a DHS. The Pearson's correlation coefficient is shown on the top

identified 76,495 DHSs with predicted log2 fold change $|\delta|>1$. Most (97%) of these differential DHSs were located outside promoter regions ($\pm 1$ kb from transcription start sites (TSS)), representing a slightly higher propensity to occur outside promoters compared to random DHSs (Supplementary Fig. 33, 97% vs. 92%, $p$-value $<10^{-15}$ by one-sided Fisher's exact test).

For evaluation, we randomly sampled 26 DHSs at different predicted fold change levels (12, 10, and 4 DHSs with $|\delta|>2$, $1 <|\delta|\leq 2$, and $0.1 <|\delta|\leq 1$, respectively) along with five non-differential DHSs ($|\delta|<0.1$) as controls. As most of the DHSs were located outside promoter regions, we conducted ChIP-qPCR

on histone modification H3K4me1 to test the predictions. High H3K4me1 level is known to be associated with active enhancers, whereas differential H3K4me1 has been observed previously in both enhancers and promoters[34]. For DHSs with predicted log2 fold change $|\delta|>2$, $1<|\delta|\leq 2$, and $0.1 <|\delta|\leq 1$, ChIP-qPCR detected differential H3K4me1 levels in 7/12, 8/10 and 1/4 cases respectively, compared to 0/5 for control DHSs (Fig. 7a, b; Supplementary Fig. 34; Supplementary Data 8). The predicted DH difference correlated well with the ChIP-qPCR measured H3K4me1 difference (Fig. 7c, Pearson's correlation $= 0.67$, Spearman's rank correlation $= 0.61$). On the basis of ChIP-qPCR, using

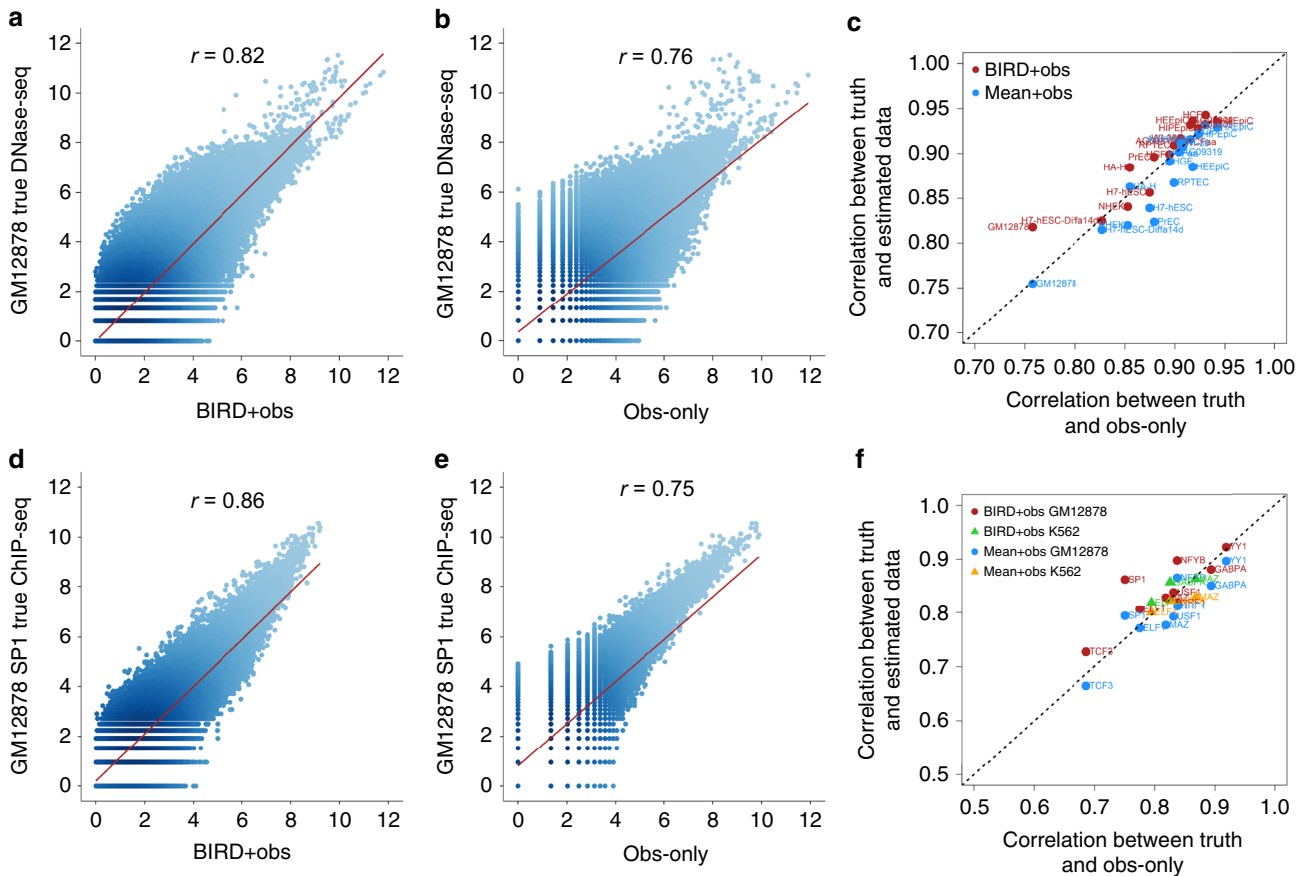

**Fig. 8** BIRD predictions used as pseudo-replicates to improve DNase-seq and ChIP-seq data analyses. The observed signal from one sample ("obs-only") in a test cell type was combined with BIRD predictions to produce the integrated signal ("BIRD + obs"). Signals before and after integration are compared with the observed signal from another sample from the same cell type ("truth"). **a–c** DNase-seq. **a** Correlation (r) between the "truth" and "BIRD + obs" (i.e., the integrated signal) in GM12878. Each dot is a genomic locus. **b** Correlation between the "truth" and "obs-only" (i.e., the original signal without integrating BIRD) in GM12878. **c** The same analyses were done for 16 test cell types. Red dots in the scatterplot compare the P–T correlation r for BIRD + obs vs. r for obs-only in the 16 cell types. BIRD + obs outperformed obs-only in 12 of 16 test cell types. As a control, BIRD was replaced by the mean DH profile of training cell types. Blue dots show the P–T correlation r for mean + obs vs. r for obs-only in the 16 test cell types. Mean + obs did not improve over obs-only. **d–f** ChIP-seq. **d** Correlation between the "truth" and "BIRD + obs" for SP1 in GM12878 at SP1 motif sites. **e** Correlation between the "truth" and "obs-only" for SP1 in GM12878 at SP1 motif sites. **f** The same analyses were done for nine TFs in GM12878 (circles) and three TFs in K562 (triangles). Once again, BIRD + obs outperformed obs-only in 9 of 12 cases (red and green), but mean + obs did not improve over obs-only (blue and yellow)

$|\delta| > 1$ as the cutoff yielded a good empirical validation rate (15/22 = 68%). The 76,495 differential DHSs are generated using this cutoff. They represent a significant amount of new information generated by BIRD.

The differential DHSs predicted by BIRD were enriched in flanking regions of differentially expressed genes, and differential genes associated with differential DHSs were enriched in neuron development and neuron differentiation functions (Supplementary Methods; Supplementary Fig. 35; Supplementary Data 9). DNA motifs enriched in DHSs downregulated in iPSC-derived neurons were linked to TFs closely involved in stem cell maintenance, such as SOX2, OCT4 (aka POU5F1), KLF4, and NANOG (Supplementary Data 10). For DHSs upregulated in iPSC-derived neurons, enriched motifs contained TFs involved in neuronal differentiation. These include important regulators of neurogenesis and neural development such as NEUROG2, NEUROD2, and ZBTB18 (aka RP58)[33, 35], as well as ATOH1, a TF recently reported to have an important role in the differentiation of dopaminergic neurons[36] (Supplementary Data 10).

Altogether, this analysis demonstrates how BIRD can be used in practice to predict differential regulatory signals. When experimental regulome data are not available, the predicted differential DHSs and their potential regulators suggested by the enriched motifs may be used as candidates to guide follow-up functional experiments (e.g., knock-out experiments) to accelerate the study of regulatory circuitry.

**Prediction as pseudo-replicate to improve regulome analysis.** In applications of high-throughput regulome profiling technologies, it is common to encounter data with low signal-to-noise ratio (SNR) or a small replicate number. Both can lead to low-signal detection power. However, if one has gene expression data, BIRD predictions may be used as pseudo-replicates to enhance the signal. To demonstrate, we analyzed DNase-seq data for GM12878 generated by ENCODE. The data had two replicates. We reserved one replicate as "truth" and used the other one as the "observed" data. Applying the BIRD prediction models trained using the 40 training cell types (GM12878 not included), we predicted DH in GM12878 and treated the prediction as a pseudo-replicate. We then estimated "true" DH using either the "observed" data alone (obs-only) or the average of the "observed" data and pseudo-replicate (BIRD + obs). After adding the pseudo-

replicate, the correlation between the predicted and true DH increased (Fig. 8a, b, $r_L$ for BIRD + obs vs. obs-only = 0.82 vs. 0.76). Replacing BIRD predictions with the mean DH profile of 40 training cell types in this analysis (mean + obs) did not yield similar increase in the P–T correlation ($r_L = 0.76$). We carried out the same analyses on 16 test cell types, and BIRD predictions improved signal in 12 of them (Fig. 8c; Supplementary Methods). Similarly, we tested whether the predicted DH can boost ChIP-seq signals using ChIP-seq data for nine TFs in GM12878 and three TFs in K562 (Supplementary Methods). BIRD improved signal in 9/12 cases (Fig. 8d–f). In Supplementary Note 5, Supplementary Figs. 36–42, and Supplementary Data 11, we show that one may also use the correlation between the BIRD-predicted pseudo-replicates and experimental DNase-seq data to check data quality. Collectively, these results demonstrate that predictions can serve as a bridge to integrate expression and regulome data.

## Discussion

In summary, this study for the first time examined systematically to what extent regulatory element activities can be predicted by gene expression alone. This is a problem with a large number of predictors and responses. We developed BIRD for this big data prediction problem. The study also demonstrates the feasibility of using gene expression to predict TFBSs and differential regulatory element activities, applying BIRD to GEO to expand the current regulome catalog, and using predictions to facilitate data integration. BIRD is a novel approach that can be used to extract information from gene expression data to study regulome. In the absence of experimental regulome data (e.g., ChIP-seq or DNase-seq data), BIRD predictions can provide valuable information to guide hypothesis generation, target prioritization, and design of follow-up experiments. When experimental regulome data are available, BIRD predictions can also serve as pseudo-replicates to improve the data analysis. Although predictions in this study were made using exon arrays, BIRD is a general approach and can be applied to other types of gene expression data when training data are available. For instance, in a separate study we observe that applying BIRD to RNA-seq allows one to predict genome-wide chromatin accessibility not only for bulk samples but also for RNA-seq samples generated using small number of cells[37].

Our results have important practical implications for the analysis of existing and future gene expression data. Conventionally, gene expression data are mainly collected to study transcriptome. The method and software developed in this study now allow one to conveniently utilize such data to study gene regulation. By adding a new component to the standard analysis pipeline of expression data, expression-based regulome prediction can bring added value to an enormous number of new and existing gene expression experiments. Given the wide application of gene expression profiling, this will greatly impact how expression data are most effectively used.

In our analyses, prediction models were trained for DHSs found in the training data. Thus, one limitation of BIRD is that it will not discover new locations of DHSs when applied to analyze a new gene expression sample. However, the number of cell types with both regulome and expression data continues to increase. As more training data become available in the near future, one can expect that most DHSs in the genome will be covered by the training data, and new DHSs uniquely present in a new sample will account for only a small fraction of the regulome (Supplementary Note 6; Supplementary Fig. 43). Importantly, knowing the genomic locations of cis-regulatory elements does not mean that activities of each element in all biological contexts are known. In this regard, BIRD has its unique advantages compared with the conventional regulome-mapping technologies. As gene

expression profiling experiments are more widely conducted than regulome-mapping experiments, the number of biological contexts with available gene expression data is orders of magnitude larger than that with experimental regulome data. BIRD can be readily applied to massive amounts of existing and new gene expression data to generate regulome information for a large number of biological contexts without experimental regulome data. In the near future, no other experimental regulome mapping technology can achieve similar level of comprehensiveness in terms of biological context coverage.

Our current study may be extended in multiple directions in the future. For instance, it is important to extend BIRD to other gene expression platforms. It also remains to be answered whether gene expression can be similarly used to predict other functional genomic data types.

## Methods

**DNase-seq data processing**. The bowtie[38] aligned (alignment based on hg19) DNase-seq data for 57 human cell types with normal karyotype were downloaded from the ENCODE in bam format (download link: http://hgdownload.cse.ucsc.edu/goldenPath/hg19/encodeDCC/wgEncodeUwDnase). The human genome was divided into 200 base pair (bp) non-overlapping bins. The number of reads falling into each bin was counted for each DNase-seq sample. To adjust for different sequencing depths, bin read counts for each sample $i$ were first divided by the sample's total read count $N_i$ and then scaled by multiplying a constant $N$ ($N = \min\{N_i\} = 17,002,867$, which is the minimum sample read count of all samples). After this procedure, the raw read count $n_{li}$ for bin $l$ and sample $i$ was converted into a normalized read count $\tilde{n}_{li} = Nn_{li}/N_i$. The normalized read counts from replicate samples were averaged to characterize the DH level for each bin in each cell type. The DH level was then log2 transformed after adding a pseudocount of 1. The transformed data were used for training and testing prediction models, treating each bin as a genomic locus. As chromosome Y was not present in all samples, we excluded this chromosome from our subsequent analyses.

**Gene expression data processing**. The Affymetrix Human Exon 1.0 Array (i.e., exon array) data for the same 57 ENCODE cell types were downloaded from GEO (GEO accession number: GSE19090). In addition, we downloaded 2000 exon array samples from GEO for constructing the PDDB database (GEO accession numbers for these samples are available at PDDB). All samples were processed using the GeneBASE[39] software to compute gene-level expression. The output of Gene-BASE was expression levels of 18,524 genes in each sample. The GeneBASE gene expression levels were log2 transformed after adding a pseudocount of 1 and then quantile normalized[40] across samples. For the 57 ENCODE cell types, replicate samples within each cell type were averaged and the averaged mean expression profile of each cell type was used for training and testing the prediction models.

**Training-test data partitioning and genomic loci filtering**. The 57 ENCODE cell types were randomly partitioned into a training dataset with 40 cell types and a test dataset with 17 cell types (Supplementary Data 1, partition #1). As not all genomic loci are regulatory elements, we first screened for genomic loci with unambiguous DH signal in at least one cell type in the training data as follows. Genomic bins with normalized read count >10 in at least one cell type were identified and retained, and the other genomic bins were excluded. Among the retained loci, bins with normalized read count >10,000 in any cell type were considered abnormal and these bins were also excluded from subsequent analyses. Finally, for each remaining bin, a SNR was computed in each cell type, and bins with small SNR in all cell types were filtered out. To compute SNR of a genomic bin in a cell type, we first collected 500 bins in the neighborhood of the bin in question. Then, we computed the average DH level of these bins. Next, the DH level was log2 transformed after adding a pseudocount of 1 to serve as the background. The $\log_2(\text{SNR})$ was defined as the difference between the normalized and log2 transformed DH level of the bin in question and the background. Genomic bins with $\log_2(\text{SNR}) > 2$ in at least one cell type were identified and retained for subsequent analyses, and the other genomic bins were excluded. After applying this filtering procedure to the 40 training cell types, 912,886 genomic bins were retained and used for training and testing prediction models for results presented in Figs. 2 and 3. Bins selected by this procedure were referred to as DNase I hypersensitive sites (DHSs) in this article. We note that the above filtering procedure only uses the training cell types. This allows one to objectively evaluate the prediction performance in real applications where models trained using the training cell types are applied to make predictions in new cell types for which DNase-seq data are not available.

To evaluate the robustness of our conclusions, we repeated the same random partitioning procedure five times, resulting in five different training-test data partitions (Supplementary Data 1). For each partition, genomic loci were filtered using the same protocol described above, and the retained loci (which depend on the training data and therefore are different for different partitions) were used to

train and test BIRD. Results from the first partition were presented in the main article, and results from the other four random partitions were similar (Supplementary Fig. 12).

Prediction models were retrained using all 57 ENCODE cell types as training data for predicting TFBSs in K562, analyzing 2000 GEO exon array samples used for constructing PDDB, and predicting differential DH in dopaminergic neuron differentiation. Applying the genomic loci filtering protocol described above to these 57 cell types resulted in 1,108,603 genomic bins for which prediction models were constructed and evaluated.

**Notations and problem formulation**. For a biological sample, let $Y_l$ be the DH level of genomic locus $l$ ($= 1, \ldots, L$), and let $X_g$ be the expression level of gene $g$ ($= 1, \ldots, G$). The genome-wide DH profile and gene expression profile are represented by two vectors $\mathbf{Y} = (Y_1, \ldots, Y_L)^{\mathrm{T}}$ and $\mathbf{X} = (X_1, \ldots, X_G)^{\mathrm{T}}$ respectively. Here, the superscript T indicates matrix or vector transpose. Both the DH and gene expression profiles are assumed to be normalized and at log2 scale. Our goal is to use $\mathbf{X}$ to predict $\mathbf{Y}$. This can be formulated as a problem of building a regression $Y_l = f_l(\mathbf{X}) + \epsilon_l$ for each genomic locus. Here $\epsilon_l$ represents random noise, and $f_l(.)$ is the function that describes the systematic relationship between the DH level of locus $l$ (i.e., $Y_l$) and the gene expression profile (i.e., $\mathbf{X}$).

The function $f_l(\mathbf{X})$ is unknown. We train it using $\mathbf{X}$ and $\mathbf{Y}$ observed from a number of different cell types. The training data are organized into two matrices: a gene expression matrix $\mathbb{X} = (x_{gc})_{G \times C}$ and a DH matrix $\mathbb{Y} = (y_{lc})_{L \times C}$. Rows in these matrices are genes and genomic loci, respectively. Columns in these matrices are cell types. $C$ is the number of training cell types. Each column of $\mathbb{X}$ and $\mathbb{Y}$ is a realization of the random vector $\mathbf{X}$ and $\mathbf{Y}$ in a specific cell type. Building the prediction model for each locus $l$ is a challenging high-dimensional regression problem as the dimensionality of the predictor $\mathbf{X}$ is much bigger than the sample size of the training data (i.e., $G \gg C$). What makes this problem even more challenging than the conventional high-dimensional problems in statistics is that one needs to solve a massive number of such high-dimensional regression problems (one for each locus) simultaneously. Thus, it is important to consider both statistical efficiency and computational efficiency when developing solutions.

In subsequent sections, various methods for training $f_l(\mathbf{X})$ will be described. Each method has a training component and prediction component. Before training prediction models, we standardize each row of $\mathbb{X}$ and $\mathbb{Y}$ in the training data to have zero mean and unit standard deviation (SD). More precisely, each DH value in $\mathbb{Y}$ is standardized using $\tilde{y}_{lc} = (y_{lc} - a_l^y)/s_l^y$, where $a_l^y$ and $s_l^y$ are the mean and SD of the DH signals at locus $l$ (i.e., row $l$ of $\mathbb{Y}$). Similarly, each expression value in $\mathbb{X}$ is standardized using $\tilde{x}_{gc} = \left(x_{gc} - a_g^x\right)/s_g^x$, where $a_g^x$ and $s_g^x$ are the mean and SD of the gene expression for gene $g$ (i.e., row $g$ of $\mathbb{X}$). The prediction models are then constructed using the standardized values $\tilde{\mathbb{X}}$ and $\tilde{\mathbb{Y}}$.

Once the models are constructed using the training data, they can be applied to new samples to make predictions. To do so, the expression profile $\mathbf{X}$ of the new sample is first quantile normalized to the quantiles of the training exon array data. The log2-transformed expression value of each gene $X_g$ in the new sample is then standardized using $\tilde{X}_g = \left(X_g - a_g^x\right)/s_g^x$, where $a_g^x$ and $s_g^x$ are the pre-computed mean and SD of the gene expression for gene $g$ in the training data. After applying the trained model to the standardized gene expression profile $\tilde{\mathbf{X}}$ to make predictions, the predicted DH value for each locus, $\tilde{Y}_l$, is transformed back using $\hat{Y}_l = s_l^y * \tilde{Y}_l + a_l^y$, where $a_l^y$ and $s_l^y$ are the pre-computed mean and SD of the DH signals for locus $l$ in the training data. The unstandardized $\hat{Y}_l$ gives the prediction for $Y_l$, the DH level of genomic locus $l$ in the new sample.

**Measures for method evaluation**. In order to evaluate prediction performance of a prediction method, the method can be applied to a number of test cell types to predict their DH profiles based on their gene expression profiles. Let $\hat{y}_{lm}$ be the predicted DH level of locus $l$ in test cell type $m$ ($= 1, \ldots, M$), and let $y_{lm}$ be the true DH level measured by DNase-seq (both are at log2 scale). Three performance statistics were used in this study (Fig. 2a):

(1) Cross-locus correlation ($r_L$). This is the Pearson's correlation between the predicted signals $\hat{\mathbf{y}}_{*m} = \left(\hat{y}_{1m}, \ldots, \hat{y}_{Lm}\right)^{\mathrm{T}}$ and the true signals $\mathbf{y}_{*m} = (y_{1m}, \ldots, y_{Lm})^{\mathrm{T}}$ across different loci for each test cell type $m$. The cross-locus correlation measures the extent to which the DH signal within each cell type can be predicted.

(2) Cross-cell-type correlation ($r_C$). This is the Pearson's correlation between the predicted signals $\hat{\mathbf{y}}_{l*} = (\hat{y}_{l1}, \ldots, \hat{y}_{lM})$. and the true signals $\mathbf{y}_{l*} = (y_{l1}, \ldots, y_{lM})$. across different cell types for each locus $l$. The cross-cell-type correlation measures how much of the DH variation across cell types can be predicted.

(3) Squared prediction error ($\tau$). This is measured by the total squared prediction error scaled by the total DH data variance in the test dataset: $\tau = \frac{\sum_l \sum_m \left(y_{lm} - \hat{y}_{lm}\right)^2}{\sum_l \sum_m \left(y_{lm} - \overline{y}\right)^2}$, where $\overline{y}$ is the mean of $y_{lm}$ across all DHSs and test cell types.

In addition to Pearson's correlation, we also computed Spearman's rank correlation as the correlation measure and obtained similar results (Supplementary

Figs. 13 and 14). For simplicity, the results based on Pearson's correlation were presented in the main article unless stated otherwise.

**Prediction based on neighboring genes**. For each genomic locus $l$, the $N$ closest genes were identified (gene annotation based on RefSeq genes of human genome hg19 downloaded from UCSC genome browser: http://hgdownload.cse.ucsc.edu/goldenPath/hg19/database/refFlat.txt.gz). The closeness was defined by the distance between the gene's transcription start site and the locus center. Using the selected genes $(\tilde{X}_{l_1}, \ldots, \tilde{X}_{l_N})$ as predictors, a multiple linear regression $\tilde{Y}_l = \beta_{l0} + \beta_{l1}\tilde{X}_{l_1} + \cdots + \beta_{lN}\tilde{X}_{l_N} + \epsilon_l$ is fit. On the basis of the fitted model, the standardized DH level of locus $l$ in a new sample is predicted using $\tilde{Y}_l = f_l(\tilde{\mathbf{X}}) = \beta_{l0} + \beta_{l1}\tilde{X}_{l_1} + \cdots + \beta_{lN}\tilde{X}_{l_N}$. We tested different values of $N$ ($=1, 2, \ldots, 20$) on a randomly selected set of DHSs ($n = 9128$; ~1% of the 912,886 DHSs obtained from the 40 training cell types). The performance for the neighboring gene approach shown in Fig. 2b and c was based on the performance achieved at the optimal $N$. For instance, Supplementary Fig. 2a shows the $r_C$ distribution for different $N$ based on the 9128 DHSs. At $N = 15$, the mean $r_C$ reached its maximum. Correspondingly, the $r_C$ distribution shown in Fig. 2c was based on $N = 15$.

We also tested whether nonlinear regression can improve the prediction. Generalized additive model with smoothing splines (GAM) were applied (using the R package "gam"[41]) to the same 1% of DHSs. However, the best prediction performance of GAM was worse than the best prediction performance of the linear regression (Supplementary Fig. 2a, see the best performance of GAM achieved at $N = 17$ vs. the best performance of linear model achieved at $N = 15$). This indicates that using non-linear model did not improve prediction accuracy. Moreover, the computational time required by GAM was substantially longer than linear regression (Supplementary Fig. 2b), making it difficult to apply to the whole genome. On the basis of this, linear regression was used to perform our genome-wide analysis.

**BIRD**. BIRD predicts DH at each genomic locus $l$ by combining predictions from two types of models, locus-level model and pathway-level model, through model aggregation. Details are provided below.

**Locus-level model $\mathrm{BIR}(\overline{\mathbf{X}}, \mathbf{Y})$**. $\mathrm{BIR}(\overline{\mathbf{X}}, \mathbf{Y})$ stands for "big data regression using clustered predictor X and original response Y". It is the basic building block of BIRD. This locus-level model begins with grouping-correlated genes into clusters. This is achieved by clustering rows of the standardized training data matrix $\tilde{\mathbb{X}}$ into $K$ clusters using $k$-means clustering[42] (Euclidean distance used as similarity measure). On the basis of the clustering result, the gene expression profile $\tilde{\mathbf{X}}$ of each sample is converted into a lower dimensional vector $\overline{\mathbf{X}} = (\overline{X}_1, \ldots, \overline{X}_K)$, where $\overline{X}_k$ is the mean expression level of genes in cluster $k$. BIRD will use gene clusters' mean expression $\overline{\mathbf{X}}$ instead of the expression of individual genes $\tilde{\mathbf{X}}$ as predictors to build prediction models. Clustering serves multiple purposes. It reduces the dimension of the predictor space. By combining correlated genes, it also reduces the co-linearity among predictors. Additionally, the cluster mean is less sensitive to measurement noise and therefore can reduce the impact of measurement error of a gene on the prediction.

After clustering, the G×C matrix $\tilde{\mathbb{X}}$ is converted into a K×C matrix $\overline{\mathbb{X}}$ ($G \approx 10^4$, $K \approx 10^2 \sim 10^3$). The predictor dimension is reduced, but it is still high compared to sample size. Borrowing the idea from recent high-dimensional regression literature[43], we further reduce the predictor dimension using a fast variable screening procedure: for each DHS locus $l$, the Pearson's correlation between its DH signal (i.e., row $l$ of $\tilde{\mathbb{Y}}$) and the expression of each gene cluster $k$ (i.e., row $k$ of $\overline{\mathbb{X}}$) across the training cell types is computed, and the top $N$ ($\approx 10^1$) clusters with the largest correlation coefficients are selected. Using the selected clusters $(\overline{X}_{l_1}, \ldots, \overline{X}_{l_N})$ as predictors, a multiple linear regression $\tilde{Y}_l = \beta_{l0} + \beta_{l1}\overline{X}_{l_1} + \cdots + \beta_{lN}\overline{X}_{l_N} + \epsilon_l$ is then fit. On the basis of the fitted model, the standardized DH level of locus $l$ in a new sample is predicted by $\tilde{Y}_l = f_l(\tilde{\mathbf{X}}) = \beta_{l0} + \beta_{l1}\overline{X}_{l_1} + \cdots + \beta_{lN}\overline{X}_{l_N}$. Of note, although each regression model only contains a small number of predictors, these predictors are selected after examining information from all genes. Therefore, training the prection model utilizes information from all genes.

$\mathrm{BIR}(\overline{\mathbf{X}}, \mathbf{Y})$ has two parameters: the cluster number $K$ and the predictor number $N$. In this study, we set $K = 1500$ and $N = 7$. These parameters were chosen based on testing different values of $K$ and $N$ ($K = 100, 200, 500, 1000, 1500, 2000$; $N = 1, 2, 3, 4, 5, 6, 7, 8$) using a fivefold cross-validation conducted within the 40 training cell types (i.e., the same training cell types used for Figs. 2 and 3) on a random subset of genomic loci (1% of all DHSs). As cross-cell-type prediction is more difficult than cross-locus prediction, we identified the optimal parameter combination as the one that maximizes the mean cross-cell-type correlation $r_C$. Supplementary Fig. 3a shows that the optimal combination was $K = 1500$ and $N = 7$. This parameter combination was then used in all subsequent $\mathrm{BIR}(\overline{\mathbf{X}}, \mathbf{Y})$, $\mathrm{BIR}(\overline{\mathbf{X}}, \overline{\mathbf{Y}})$, and BIRD models throughout this study.

In $\mathrm{BIR}(\overline{\mathbf{X}}, \mathbf{Y})$, clustering co-expressed genes is an important step since it improves prediction performance. In fact, Supplementary Figs. 3 and 4 compare $\mathrm{BIR}(\overline{\mathbf{X}}, \mathbf{Y})$ with a modified version of $\mathrm{BIR}(\overline{\mathbf{X}}, \overline{\mathbf{Y}})$ that skips the clustering step. This modified version, denoted by $\mathrm{BIR}(\mathbf{X}, \mathbf{Y})$, uses individual genes rather than gene clusters as predictors. It is a special case of $\mathrm{BIR}(\overline{\mathbf{X}}, \mathbf{Y})$ when the gene cluster number $K$ is equal to the gene number $G$. $\mathrm{BIR}(\mathbf{X}, \mathbf{Y})$ was not used in BIRD, but the comparison between $\mathrm{BIR}(\mathbf{X}, \mathbf{Y})$ and $\mathrm{BIR}(\overline{\mathbf{X}}, \mathbf{Y})$ allowed us to study the effect of

gene clustering on prediction. $\mathrm{BIR}(X, Y)$ only has one parameter: the number of predictors $N$. We first compared $\mathrm{BIR}(X, Y)$ and $\mathrm{BIR}(\overline{X}, Y)$ ($K = 1500$) when both methods used the same $N$ and found that $\mathrm{BIR}(\overline{X}, Y)$ consistently outperformed $\mathrm{BIR}(X, Y)$ (Supplementary Fig. 3b). Next, based on fivefold cross-validation performed on the 40 training cell types using 1% of all DHSs from these training cell types, we identified $N = 5$ as the optimal value for $\mathrm{BIR}(X, Y)$ (Supplementary Fig. 3a). We then compared $\mathrm{BIR}(X, Y)$ based on this optimal $N$ ($N = 5$) to $\mathrm{BIR}(\overline{X}, Y)$ with its optimal parameter ($K = 1500$ and $N = 7$) in Supplementary Fig. 4. $\mathrm{BIR}(\overline{X}, Y)$ again outperformed $\mathrm{BIR}(X, Y)$ by producing higher P–T correlations and lower squared prediction error.

In Supplementary Note 1 and Supplementary Figs. 5 and 6, we further compared $\mathrm{BIR}(\overline{X}, Y)$ with a number of alternative prediction methods including lasso[44], fused lasso[15], group lasso[16], composite minimax concave penal regression (composite MCP)[17], linear regression with stepwise predictor selection[45] (SPS), $k$-nearest neighbors[46] (KNN), random forest[47] (RF), and principal component regression[48] (PCR). This benchmark analysis shows that $\mathrm{BIR}(\overline{X}, Y)$ not only offers the best prediction accuracy but also is computationally efficient. On the basis of this result, $\mathrm{BIR}(\overline{X}, Y)$ was used as the basic building block for subsequent modeling.

**Pathway-level model $\mathrm{BIR}(\overline{X}, \overline{Y})$.** $\mathrm{BIR}(\overline{X}, \overline{Y})$ stands for "big data regression using clustered predictor X and clustered response Y". In addition to clustering co-expressed genes, $\mathrm{BIR}(\overline{X}, \overline{Y})$ also groups genomic loci with similar DH patterns into clusters. This is done by clustering rows of the standardized matrix $\widetilde{\mathbb{Y}}$ into $H$ clusters using $k$-means clustering (Euclidean distance used as similarity measure). Each cluster of genomic loci is viewed as a pathway. Based on the clustering result, the DH profile $\widetilde{Y}$ of each sample can be converted into a lower dimensional vector $\overline{Y} = (\overline{Y}_1, \ldots, \overline{Y}_H)$, where $\overline{Y}_h$ is the mean DH level of DHSs in cluster $h$. Instead of predicting the DH level $\widetilde{Y}$ of individual loci, $\mathrm{BIR}(\overline{X}, \overline{Y})$ uses the cluster-level gene expression $\overline{X}$ to predict cluster-level DH $\overline{Y}$ (also called pathway activities). The prediction models are constructed using linear regression in a way similar to how the regression models are constructed in $\mathrm{BIR}(\overline{X}, Y)$. In Supplementary Fig. 7, the pathway-level model $\mathrm{BIR}(\overline{X}, \overline{Y})$ was compared with the locus-level model $\mathrm{BIR}(\overline{X}, Y)$ to illustrate that cluster-level DH (i.e., pathway activities) can be predicted with higher accuracy than DH at individual genomic loci. The same parameter combination $K = 1500$ and $N = 7$ was set for both $\mathrm{BIR}(\overline{X}, Y)$ and $\mathrm{BIR}(\overline{X}, \overline{Y})$. For $\mathrm{BIR}(\overline{X}, \overline{Y})$, $H$ was set to 1000, 2000, and 5000, respectively.

**Model aggregation.** To produce the final prediction for a genomic locus, BIRD combines the locus-level and pathway-level models through a weighted average. The rationale is as follows. $\mathrm{BIR}(\overline{X}, Y)$ is a special case of $\mathrm{BIR}(\overline{X}, \overline{Y})$ when DHSs are not clustered (i.e., $H = L$). As using the cluster mean can reduce the variance of measurement noise, pathway activities are expected to be less noisy than locus-level DH measurements. As a result, $\mathrm{BIR}(\overline{X}, \overline{Y})$ was able to predict pathway activities much more accurately than using the locus-level model $\mathrm{BIR}(\overline{X}, Y)$ to predict DH levels of individual loci (Supplementary Fig. 7). The pathway activity predicted by $\mathrm{BIR}(\overline{X}, \overline{Y})$ can also serve as a prediction of the DH level for each individual locus within the pathway. This locus-level prediction may be biased, but it is usually associated with smaller variance. By contrast, predictions by $\mathrm{BIR}(\overline{X}, Y)$ for each locus may be less biased but has larger variance. By combining these two types of predictions, one may improve the overall locus-level prediction accuracy via a better tradeoff between the prediction bias and variance.

BIRD adopts this idea and implements it by combining multiple $\mathrm{BIR}(\overline{X}, \overline{Y})$ models with different $H$ values through model averaging. Consider making predictions for a sample. Let $\mathcal{H}$ be the set of $H$ values used in $\mathrm{BIR}(\overline{X}, \overline{Y})$. In this study, $\mathcal{H} = \{1000, 2000, 5000, L\}$. For each DHS locus $l$, let $\hat{Y}_l^{(H)}$ denote the locus-level DH predicted by $\mathrm{BIR}(\overline{X}, \overline{Y})$ using cluster number $H$. $\hat{Y}_l^{(L)}$ represents the locus-level DH predicted by $\mathrm{BIR}(\overline{X}, Y)$. The final locus-level DH prediction by BIRD for locus $l$ is a weighted average

$$\frac{\sum_{H \in \mathcal{H}} d_l^H \hat{Y}_l^{(H)}}{\sum_{H \in \mathcal{H}} d_l^H},$$

where $d_l^H$ is the weight. For given cluster number $H$, the weight $d_l^H$ is determined using training data as follows. Let $\tilde{\mathbf{y}}_l = (\tilde{y}_{l1}, \ldots, \tilde{y}_{lM})$ be the standardized locus-level DH for locus $l$ observed in $M$ training cell types. Each locus $l$ is associated with a cluster. Let $\tilde{\mathbf{y}}_l^{(H)} = \left(\tilde{y}_{l1}^{(H)}, \ldots, \tilde{y}_{lM}^{(H)}\right)$ represent the average of the standardized DH level of all loci within the cluster corresponding to locus $l$ in the $M$ training cell types. Define $d_l^H$ as the Pearson's correlation between the two vectors $\tilde{\mathbf{y}}_l^{(H)}$ and $\tilde{\mathbf{y}}_l$. Note that when $H = L$, $\mathrm{BIR}(\overline{X}, \overline{Y})$ reduces to $\mathrm{BIR}(\overline{X}, Y)$, and we have $\tilde{\mathbf{y}}_l^{(L)} = \tilde{\mathbf{y}}_l$ and $d_l^L = 1$. Thus, the weight for $\mathrm{BIR}(\overline{X}, Y)$ is 1.

Comparisons between the final BIRD prediction for individual genomic loci and the locus-level prediction by $\mathrm{BIR}(\overline{X}, Y)$ in Supplementary Fig. 4 show that model aggregation (i.e., BIRD) consistently improved the locus-level DH prediction compared to $\mathrm{BIR}(\overline{X}, Y)$. Therefore, the aggregated model is used as our final BIRD prediction model for each genomic locus.

**Random prediction models by permutation.** To construct random prediction models, we permuted the cell type labels of DNase-seq data in the training dataset. This permutation broke the connection between DNase-seq and gene expression data. BIRD was then trained using the permuted training dataset, and the trained model was applied to predict DH in the test dataset. The permutation was performed 10 times. The statistics $r_L$, $r_C$, and $\tau$ were computed to evaluate the prediction performance of each permutation. The average values of these three statistics from the 10 permutations were used to represent the prediction performance of random prediction models.

**Wilcoxon signed-rank test for comparing different methods.** In order to compare the prediction accuracy of each pair of methods in Fig. 2b and c, a two-sided Wilcoxon signed-rank test was performed to obtain $p$-values. For instance, to test whether two methods A and B perform equally in terms of $r_L$, the paired $r_L$ values from these two methods for each cell type was obtained. Then the $r_L$ pairs from all cell types are used for the Wilcoxon signed-rank test. Similarly, to compare methods A and B in terms of $r_C$, the paired $r_C$ values for each locus was obtained, and $r_C$ pairs from all genomic loci were used for the Wilcoxon signed-rank test. $p$-values $< 10^{-4}$ were marked with "*" and $p$-values $< 10^{-15}$ were marked with "**" in Fig. 2b and c. We did not perform similar test for the squared prediction error ($\tau$) since there is only one $\tau$ for each method.

**Predicting differential DH.** In order to evaluate the ability of a method to predict differential DH signals between two sample types, we first computed the difference in predicted DH value (at log2 scale) between two cell types at each locus. We then computed the Pearson's correlation between the predicted DH difference and true DH difference (determined by DNase-seq) for each pair of test cell types. The analysis was applied to all DHSs and differential DHSs respectively. The differential DHSs were obtained by first filtering out loci without significant DH signals (defined as log2 DH level smaller than 2) in both cell types and then collecting all remaining DHSs with |True DH difference (at log2 scale)| between the two compared cell types |>1. To investigate how the prediction performance depends on the similarity level of the two compared cell types, we calculated the similarity between the cell types using the Pearson's correlation of their DH profiles (i.e., true DH levels across all loci). We then grouped all pairs of cell types into four strata based on the quartiles of the correlation coefficients. Figure 3g shows the prediction performance for each stratum. We also used the pathway-level model to predict differential pathway activities and evaluated the pathway-level prediction performance in a similar fashion (Fig. 3g; Supplementary Fig. 11).

**Construction of the PDDB database.** BIRD prediction models trained using the 57 ENCODE cell types were applied to predict DH levels at the 1,108,603 genomic loci for 2000 human exon array samples obtained from GEO. Each GEO sample in PDDB was annotated using its GSE number, GSM number, cell type, cell status, gender and other information such as age. The predicted DH and the annotation data were both stored in PDDB. A track data hub[49] was set up in the UCSC genome browser for visualizing the predicted DH signal.

Using the PDDB user interface (Fig. 5b), users can retrieve the predicted DH profile from user-specified genomic regions and samples. For instance, one can input a list of genomic regions (Fig. 5a, "Step 1") and enter a keyword such as "stem cell" in the "Cell Type" searching field (Fig. 5a, "Step 2"). After clicking the "Search" button, a list of samples matching the keyword will be returned (Fig. 5b). One can then choose to download the predicted DH profile from the input genomic regions and selected samples, download sample annotation data, and visualize the predicted DH profiles in the UCSC genome browser (Fig. 5b, c). PDDB is available at http://jilab.biostat.jhsph.edu/~bsherwo2/bird/index.php.

**Protocols for other analyses and examples.** The detailed methods for the analysis of factors affecting cross-cell-type prediction accuracy, comparison of BIRD and ChromImpute, analysis of predictors selected by BIRD, predicting TFBSs, regulome prediction using public expression samples in GEO, predicting differential signals in a differentiation system, and prediction as pseudo-replicate to improve regulome analysis can be found in Supplementary Methods and Supplementary Notes.

**Code availability.** The BIRD software and its source code are available at https://github.com/WeiqiangZhou/BIRD. Models trained using the 57 ENCODE cell types have been stored in the software package. With these pre-compiled prediction models, making predictions on new samples provided by users is computationally fast. On a computer with 2.5 GHz CPU and 10 Gb RAM, it took <2 min to make predictions for ~1 million DHSs in 100 samples.

**Data availability.** Exon array data used for training and testing BIRD models are all available in GEO (accession numbers: GSE19090, GSE15805, GSE9703, GSE24976, GSE32219, and GSE93012), and the dataset or sample accession numbers used for each example are provided in "Methods" and Supplementary Methods. Accession numbers of the 2000 GEO exon array samples used for constructing PDDB are available at the PDDB website (http://jilab.biostat.jhsph.edu/~bsherwo2/bird/index.php). MYC ChIP-seq data in P493-6 cells are available in

GEO (accession number: GSE51004). SOX2 ChIP-seq data in H9 human embryonic stem cells are available in GEO (accession number: GSE46837). The other TF ChIP-seq data and DNase-seq data used in this study are available from the ENCODE (http://hgdownload.cse.ucsc.edu/goldenPath/hg19/encodeDCC), and the download links for each analysis are provided in "Methods" and Supplementary Methods.

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

## Acknowledgements

We would like to thank Drs X. Shirley Liu, Chongzhi Zang, Cliff Meyer, and Yingying Wei for insightful discussions. This research is supported by grants from the Maryland Stem Cell Research Fund (2012-MSCRFE-0135-00) and the National Institutes of Health (R01HG006282 and R01HG006841), and an IDIES seed fund from the Institute for Data Intensive Engineering and Science of the Johns Hopkins University.

## Author contributions

H.J. conceived the study. W.Z. and H.J. designed and developed the methods, analyzed the results, and wrote the manuscript. W.Z. implemented the BIRD software. B.S. implemented PDDB and contributed to comparing BIRD with other prediction algorithms. Z.J. implemented CABS and developed the TFBS prediction method using improved motif models. F.D. developed the method for clustering DHSs. J.B. contributed to the collection of DNase-seq and exon array data from the ENCODE. Y.X. and M.Y. performed ChIP-qPCR and exon array experiments for the differentiation of iPSCs to

dopaminergic neurons. All authors contributed to manuscript editing and revision and approved its final version.

## Additional information

**Competing interests:** The authors declare no competing financial interests.

