## [Peer Review file · Nature Communications]

Reviewers' comments:

Reviewer #1 (Remarks to the Author):

Summary of the key results:

This paper presents a novel method for using gene expression data to predict which regions of the genome are accessible, a task that has not been solved effectively in previous studies. This is an important problem because there are many publicly available datasets with gene expression data but no accessibility data; knowing which regions are accessible could provide important insights into the transcriptional regulation that is happening in these samples. The paper formulates this task as a regression problem by using clusters of genes with correlated expressions to predict accessibility. The paper demonstrates that incorporating expressions of all genes into their model works better than using only the expressions of the genes surrounding each region. They show that their method performs slightly better than standard techniques, and that predicting clusters of DH sites produces an additional small increase in performance. They go on to show that their predicted DH sites, in conjunction with transcription factor (TF) binding motifs, can be used to predict TF binding in-vivo. In addition, when run on datasets without available accessibility data, the model provided predictions that matched known biology.

Originality and interest: (if not novel, please give references:)

Grouping highly correlated predictors to improve prediction performance in a $p > n$ regression has been extensively studied in the statistics literature (see, for example, A Selective Review of Group Selection in High-Dimensional Models). The most popular method is the fused lasso (Sparsity and smoothness via the fused lasso), which has been used to analyze gene expression data in various settings. To my knowledge, this is the first time that a group selection method has been used to predict DH sites from gene expression. The ability to predict chromatin accessibility from gene expression data would be incredibly useful if it allowed researchers to reduce the required number of accessibility experiments required to confidently identify DH sites, but the method presented in this manuscript is probably not powerful enough to allow accessibility experiments to be replaced. If the authors were able to show that their method was able to facilitate precise prediction of DH sites in novel cell types, then it would be much more interesting.

Data & methodology:

-In general, the approach and subsequent analysis were reasonable and well-presented. I've listed a few exceptions below:

-In the section titled "Cross-cell-type prediction accuracy varies greatly among different loci", the authors note that DH regions with low variance between cell types are poorly predicted by gene expression, and speculate that this is due to noisy signal in such regions. However, it also seems reasonable that regions with a low coefficient of variation are only accessible in a limited number of cell types, causing the decreased predictive power. The authors should expand this analysis to include additional ranking measures besides CV (e.g. mean accessibility score or max-min spread across samples) and/or other analyses which would help to test their noise hypothesis.

-The authors do not compare their method to any methods in the group selection literature. At the very least, they should compare to the fused lasso.

-I'm skeptical that using TF peaks that overlap JASPAR motifs as the gold-standard for TF binding sites is not overly optimistic. An alternative would be to use the reproducible/rank consistent peaks based on the irreproducible discovery rate (from encodeproject.org) as the gold-standard peak set. In addition, a comparison of the results to existing methods for this task, such as PIQ (Sherwood et al., Nature Biotechnology, 2014) and Iterative CENTIPEDE (Moyerbrailean et al., PLoS Genetics, 2016), should be added.

-The authors show that their method is superior to chromimpute "on average". The most important application of their method would be to predict DH sites in previously poorly studied cell types, but in such cell types the primary goal is typically to identify DH sites that are differentially accessible when compared to a closely related cell type (eg neural progenitor differentiation).

Unfortunately a method that appears to perform well on average may do poorly at predicting such

changes. The authors should repeat their analysis when restricted to regions that are differentially expressed in closely related cell types.

- The Results section does not include any interpretation of the model. It would be great if a paragraph could be added on a few examples of specific genes or groups of genes whose expressions are predictive of specific accessibility regions. For example, I would be interested to see if promoters' accessibilities are often predicted mostly by their corresponding gene, if enhancers with known targets' accessibilities are often predicted mostly by those targets, or highly-conserved regions' accessibilities are often predicted by genes involved in pathways that are conserved across species.

Appropriate use of statistics and treatment of uncertainties:

The analysis is generally well presented and accurate, with a couple exceptions listed below.

-The authors (mostly) use pearson correlation as their similarity measure, but they should include a ranked based measure as well.

-All of the sensitivity/predicted site graphs should be converted into the more standard sensitivity/specificity plot

Conclusions:

This method has potential to be very useful, but the results are a bit weak if the method is primarily being sold as a predictive model. As written, it confirms the expected results that gene expression is on average predictive of accessibility, and shows that clustering predictors can improve prediction accuracy. These results are interesting but the ability to predict the cell-type specific DHSs (which are arguably the most relevant and hardest to predict) accurately is not clearly demonstrated.

Suggested improvements:

This manuscript would be much more interesting if the authors were able to show that their method could be used to predict differentially accessible regions. The authors' method also needs to be compared to other methods from the group selection literature.

References:

Again, the authors need to discuss the major group selection methods.

Clarity and context:

The abstract and conclusion are accurate and clear.

Reviewer #2 (Remarks to the Author):

To the authors:

Summary: The authors develop an approach for predicting DNASE hyper sensitive (DH) sites using gene expression data. This approach is novel in that it's the first attempt to analyze these data in a reverse fashion: mRNA -> DH instead of vice versa. The author's model BIRD has several variations. In addition, the BIRD approach has carefully taken into account factors related to the computational and statistical challenges associated with processing massive datasets. They have applied their approach to multiple data scenarios and have compared with existing methods, permuted data, and compared its potential improvement in enhancing sparse or low-quality data (so called pseudo replicates).

Major comments:

1. Overall, the authors present several BIRD models. The flow/presentation of these models is in a somewhat linear fashion and is very confusing. It is still unclear to the reviewer what are the distinct advantages of each (are they all needed?). Can't one or two of these approaches be used instead of all of them?
2. It seems that overall the improvements of BIRD is fairly low. For example, the MYC example

was very low (~56%). The authors need to do a better job justifying why these results are useful in practice by better justifying the benefits of their predictions

3. Clustering: K means were used and compared with unclustered variable selection. How about basing the BIRD regression models using other dimension reduction approaches such as PCA? (e.g. use PCA components in the regression instead of K means).

4. One important analysis that seems lacking would be in BIRD's ability to identify differential DH regions from differentially expressed genes in a treatment/control or disease/no disease state. Would be very useful to see if BIRD works in this case.

Reviewer #3 (Remarks to the Author):

The paper presents BIRD, a method to predict the DNase I hypersensitivity signal of predefined genomic locations (which were detected as DHS in at least one of the training samples) based on the transcriptional activity in the whole genome. This means that it cannot discover new (previously undetected) DHS, only predict the hypersensitivity of those that are already known to be hypersensitive in some cell lines/samples. It compares the performance of several variants of the method with a simpler approach, as well as with a related existing software (ChromImpute, which is more general but requires multiple types of experiments as input, most commonly ChIP-seq for multiple histone modifications). It also claims to provide the results of the method applied to 2000 existing exon array samples as a web resource (PDDB).

The paper is reasonably well-written and the points made in the text generally well supported by figures, and the methods are clearly explained. I am yet to be convinced in its broad practical utility. Since its predictions are based on training on existing data, no new cell-type specific biology is likely to be inferred, even though the predictions based on what is already known can be useful.

Specific points:

Results, 2nd section, paragraph 3, states "transcription of a gene may be controlled by multiple cis-regulatory elements, DH of a particular regulatory element may not always correlate well with neighboring gene expression". This is not just because of multiple elements, but the fact that many elements are known to control genes over a very long genomic distance, in many cases across other genes.

The first two test statistics used in Figure 1 could be better explained, with reference to the underlying biology, as "Cross-locus correlation" and "cross cell-type correlation" are not easy to interpret at first glance, and they are important to the section, e.g. "Relative levels of DH within a cell type can be accurately predicted", and "Variation of a single DH site across cell types..." The authors do give a good explanation of the causes of the discrepancy in performance, which is convincing.

There is a brief mention of "pathway" in the description of the BIRD \bar{x} \bar{y} model, however it would be interesting if a further analysis of the clusters was given. Naively, it could be expected that each cluster in X could represent a transcription factor and its downstream targets (or multiple, closely related TFs), and Y might represent the binding sites for that TF and any downstream TFs. This could represent one of the most interesting findings in the paper.

The comparison of BIRD with ChromImpute is not entirely convincing, as in the plot in the main figure Pearson correlation of P-T values for BIRD, which are DNase levels, are compared with the correlation of P-T values for ChromImpute, which are predicted log₁₀ p-values. The authors

address this in the supplementary figures by using Spearman correlation, which may level the playing field somewhat, but while they state that "neither method consistently outperformed the other", there are several outlier cases (GM12878, Monocytes and NHEK) for which ChromImpute performs well above BIRD, and these are also the cell types with the largest deviation from the average (based on the average prediction model). The result is still important, however, given that ChromImpute is based on histone modification which may be expected to better predict DH, since H3K27ac and H3K4me3 mark active enhancers and promoters.

Prediction of TFBS using expression data is interesting, however the authors could surely find a better example than MYC - as they say, the motif is bound by many other transcription factors, and predictions at an FDR of approximately 50% are not generally useful.

The resource provided which has reanalysed 2000 GEO samples could be useful to some researchers, however it seems odd that using the website more results can be submitted using the text field than as BED coordinates ("2MB size limit. Larger entries can be entered in text area below"). Manually selecting and pasting in large queries is difficult and makes automation impossible on most systems.

The demonstrate that BIRD can sometimes improve DNase predictions based on poor replicates, but a more interesting use case would be the detection of outlier samples, as is done in the ChromImpute paper, and a similar analysis would be more convincing here in demonstrating its usefulness as a QC metric.

The absence of evidence that the method works comparably well on RNA-seq as on exon array expression data is a quite serious omission, since most expression data in the future will come from RNA-seq and related methods and not exon arrays.

BLUE: Reviewer's Questions and Comments

BLACK: Responses

RED: Revised texts in the article

Reviewer #1 (Remarks to the Author):

Summary of the key results:

This paper presents a novel method for using gene expression data to predict which regions of the genome are accessible, a task that has not been solved effectively in previous studies. This is an important problem because there are many publicly available datasets with gene expression data but no accessibility data; knowing which regions are accessible could provide important insights into the transcriptional regulation that is happening in these samples. The paper formulates this task as a regression problem by using clusters of genes with correlated expressions to predict accessibility. The paper demonstrates that incorporating expressions of all genes into their model works better than using only the expressions of the genes surrounding each region. They show that their method performs slightly better than standard techniques, and that predicting clusters of DH sites produces an additional small increase in performance. They go on to show that their predicted DH sites, in conjunction with transcription factor (TF) binding motifs, can be used to predict TF binding in-vivo. In addition, when run on datasets without available accessibility data, the model provided predictions that matched known biology.

We thank the reviewer for the comprehensive summary of our study.

Originality and interest:

Grouping highly correlated predictors to improve prediction performance in a $p > n$ regression has been extensively studied in the statistics literature (see, for example, A Selective Review of Group Selection in High-Dimensional Models). The most popular method is the fused lasso (Sparsity and smoothness via the fused lasso), which has been used to analyze gene expression data in various settings. To my knowledge, this is the first time that a group selection method has been used to predict DH sites from gene expression.

Thank you for pointing out that this is the first time that grouping correlated predictors in high-dimension has been used in our problem setting. In the revised manuscript, we have also added a discussion about the connection between our method and group variable selection:

*"Conceptually, constructing regression models using selected gene clusters can be viewed as a group variable selection approach applied in a prediction setting¹⁴. However, unlike $BIR(\bar{X}, Y)$, conventional group variable selection methods are primarily developed for modeling a univariate response using high-dimensional predictors¹⁵⁻¹⁷. They pay less attention to important issues such as computational efficiency and robustness to noisy predictors for handling complex big data where both predictors and responses are high-dimensional. We compared $BIR(\bar{X}, Y)$ with the popular group variable selection methods group lasso¹⁶ and composite minimax concave penalty regression¹⁷ (composite MCP). $BIR(\bar{X}, Y)$ was more accurate and computationally efficient than the group penalty methods. We also compared $BIR(\bar{X}, Y)$ with fused lasso¹⁵ based on 1% of the genome. Both methods yielded similar prediction accuracy, but fused lasso was $>10^5$ times slower than $BIR(\bar{X}, Y)$. A whole-genome run of $BIR(\bar{X}, Y)$ took less than one day using one CPU, whereas a whole-genome run of fused lasso under the same condition is estimated to take >500 years (**Supplementary Fig. 5, Supplementary Notes**)."*

The ability to predict chromatin accessibility from gene expression data would be incredibly useful if it allowed researchers to reduce the required number of accessibility experiments required to confidently identify DH sites, but the method presented in this manuscript is probably not powerful enough to allow accessibility experiments to be replaced. If the authors were able to show that their method was able to facilitate precise prediction of DH sites in novel cell types, then it would be much more interesting.

Thank you for appreciating the value of predicting chromatin accessibility from gene expression. We note that the purpose of this study is not to develop a method to “replace” the chromatin accessibility experiments. Instead, our goal is to develop a method “complementary” to the chromatin accessibility experiments. Therefore, we agree with the reviewer that BIRD is not used as a replacement of the chromatin accessibility experiments.

We want to emphasize that a “complementary” method does not mean that it is not valuable. In fact, due to their complementary nature, both the chromatin accessibility experiments and BIRD have their unique values and their combination can greatly accelerate the studies of gene regulation.

For mapping regulatory element activities, two problems need to be solved: (1) finding locations of all regulatory elements in a genome and (2) documenting activities of each element in all biological contexts (i.e. different cell types and conditions). Both problems are important. Unfortunately, no existing method can effectively solve both problems. The chromatin accessibility experiments are good at locating regulatory elements in a new sample, but they do not offer the efficiency required for mapping regulatory element activities in a massive number of new samples from different cell types. BIRD, on the other hand, assumes that locations of regulatory elements in the genome are given. Thus, it is not designed for finding new locations of regulatory elements in a new sample. However, given regulatory element locations, BIRD can rapidly predict regulatory element activities in a massive number of new sample types. This cannot be efficiently done by chromatin accessibility experiments. Thus, the combination of chromatin accessibility experiments and BIRD prediction can provide a realistic solution to mapping both the locations and activities of cis-regulatory elements.

In terms of predicting activities of known regulatory elements in new cell types, our evaluation of BIRD was based on using test cell types not contained in the training cell types. Thus, our results have demonstrated the ability of BIRD to predict activities of regulatory elements in new cell types. Furthermore, we also strengthened our results by adding new analyses to show BIRD’s ability to predict DH differences between different cell types, including differential DH in a differentiation system involving a new cell type not covered by ENCODE (see below). Like any other prediction method, we do not expect BIRD prediction to be perfect. However, prediction can help researchers in multiple ways to study gene regulation. Examples include but are not limited to identifying promising regulatory elements active in a sample type of interest for follow-up studies (e.g., to guide knock-out experiments), developing hypotheses regarding how a gene is controlled, and using prediction as a bridge to integrate multiple data types. As such, BIRD predictions are tremendously useful.

In terms of finding new locations of regulatory elements in new cell types, one point we want to make is that projects such as ENCODE and Roadmap Epigenomics are continuing to generate new regulome and gene expression data from new sample types. As more training data become available in the near future, one can expect that most DHS locations in the genome will be covered by the training data, and new DHSs uniquely present in a new sample will account for only a small fraction of the genome. To demonstrate this, we conducted a subsampling analysis in which we gradually increased the number of cell types in the training data and asked how many new DHS locations one can find by

adding a new cell type to the training data. Details of this analysis are described in **Supplementary Notes**, and the results are shown in **Supplementary Figure 39**. Let $Y(n)$ denote the number of DHSs discovered from n training cell types. **Supplementary Figure 39a** shows $Y(n)$ as a function of n . Mean and standard deviation from 10 simulations are shown. **Supplementary Figure 39b** shows the percentage of new DHSs contributed uniquely by adding a new cell type (i.e., $[Y(n)-Y(n-1)]/Y(n)$) as a function of n (i.e., the number of training cell types). This analysis shows that as the number of available cell types in the training data increases, the increase in the number of DHSs gradually slows down. When a few dozens of training cell types are available, new regulatory elements missed by BIRD in any new sample account for only a small fraction of the DHSs in the training data.

Supplementary Figure 39. The impact on the discovery of new DHSs by adding a new cell type to training data. (a) The number of DHSs (denoted by $Y(n)$) discovered from training data is shown as a function of the number of training cell types (denoted by n). Mean and standard deviation from 10 simulations are shown (see **Supplementary Notes**) (b) Percentage of new DHSs contributed uniquely by adding a new cell type. This plot shows $[Y(n)-Y(n-1)]/Y(n)$ as a function of n . The mean from 10 simulations is shown.

These are now discussed in the **Discussion** section and **Supplementary Notes**.

*“In our analyses, prediction models were trained for DHSs found in the training data. Thus, one limitation of BIRD is that it will not discover new locations of DHSs when applied to analyze a new gene expression sample. However, the number of cell types with both regulome and expression data continues to increase. As more training data become available in the near future, one can expect that most DHSs in the genome will be covered by the training data, and new DHSs uniquely present in a new sample will account for only a small fraction of the regulome (**Supplementary Notes, Supplementary Fig. 39**). Importantly, knowing the genomic locations of cis-regulatory elements*

does not mean that activities of each element in all biological contexts are known. In this regard, BIRD has its unique advantages compared to conventional regulome mapping technologies. Since gene expression profiling experiments are more widely conducted than regulome mapping experiments, the number of biological contexts with available gene expression data is orders of magnitude larger than the number of contexts with experimental regulome data. BIRD can be readily applied to massive amounts of existing and new gene expression data to generate regulome information for a large number of biological contexts without experimental regulome data. In the near future, no other experimental regulome mapping technology can achieve similar level of comprehensiveness in terms of biological context coverage."

Data & methodology:

In general, the approach and subsequent analysis were reasonable and well-presented. I've listed a few exceptions below:

-In the section titled "Cross-cell-type prediction accuracy varies greatly among different loci", the authors note that DH regions with low variance between cell types are poorly predicted by gene expression, and speculate that this is due to noisy signal in such regions. However, it also seems reasonable that regions with a low coefficient of variation are only accessible in a limited number of cell types, causing the decreased predictive power. The authors should expand this analysis to include additional ranking measures besides CV (e.g. mean accessibility score or max-min spread across samples) and/or other analyses which would help to test their noise hypothesis.

This is an excellent point. In the revised manuscript, we have expanded the analysis as you suggested. We found that CV, max-min spread and cell type specificity can all affect cross-cell-type prediction accuracy. Loci with low CV, low max-min spread, or highly cell-type-specific tend to have lower accuracy. We also ranked loci based on their mean accessibility, but we did not find strong correlation between mean accessibility and cross-cell-type prediction accuracy. Details of these new analyses are presented in **Methods, Figure 2a-d, and Supplementary Figures 8-9**.

Based on these new findings, we revised our conclusion and presentation in the main article as follows:

*"Multiple factors may influence cross-cell-type prediction accuracy of a locus. First, a subset of loci was not active in any test cell type. For these loci, the cross-cell-type correlation between the predicted and true DH signals (which are essentially noise) is expected to be low. After excluding these noisy loci, we found that loci with low signal range (characterized by the difference between the maximal and minimal DH values), low signal variability (characterized by coefficient of variation), or high cell-type-specificity (characterized by the number of cell types in which the locus is active or inactive) tend to have lower r_c (**Fig. 2a, Supplementary Fig. 8-9, Methods**). Together, these factors and noisy loci explained the majority (85%) of loci with low r_c (i.e., $r_c < 0.25$) (**Fig. 2b-d**). In practice, one may use these factors to screen for loci whose cross-cell-type prediction is likely to be accurate. For instance, if one filters out the noisy loci and loci with low max-min spread, low CV or high cell-type-specificity based on true DH values in the training data and predicted DH values in the test data, the mean r_c would become 0.6 (compared to the mean of 0.5 for all loci, and 0.43 for filtered loci), and 74% and 30% of loci would have $r_c > 0.5$ and >0.75 respectively (**Fig. 2e, Methods**). We also investigated the relationship between r_c and the mean DH level of a locus but did not find strong correlation between them (**Methods, Supplementary Fig. 8e-f**)."*

-The authors do not compare their method to any methods in the group selection literature. At the very least, they should compare to the fused lasso.

Thank you for this suggestion. In the revised manuscript, we compared BIRD with several popular group variable selection methods. This is described in the main article as follows:

*"Conceptually, constructing regression models using selected gene clusters can be viewed as a group variable selection approach applied in a prediction setting¹⁴. However, unlike $BIR(\bar{X}, Y)$, conventional group variable selection methods are primarily developed for modeling a univariate response using high-dimensional predictors¹⁵⁻¹⁷. They pay less attention to important issues such as computational efficiency and robustness to noisy predictors for handling complex big data where both predictors and responses are high-dimensional. We compared $BIR(\bar{X}, Y)$ with the popular group variable selection methods group lasso¹⁶ and composite minimax concave penalty regression¹⁷ (composite MCP). $BIR(\bar{X}, Y)$ was more accurate and computationally efficient than the group penalty methods. We also compared $BIR(\bar{X}, Y)$ with fused lasso¹⁵ based on 1% of the genome. Both methods yielded similar prediction accuracy, but fused lasso was $>10^5$ times slower than $BIR(\bar{X}, Y)$. A whole-genome run of $BIR(\bar{X}, Y)$ took less than one day using one CPU, whereas a whole-genome run of fused lasso under the same condition is estimated to take >500 years (**Supplementary Fig. 5, Supplementary Notes**)."*

Details of these results are presented in **Supplementary Notes** and **Supplementary Figure 5**.

-I'm skeptical that using TF peaks that overlap JASPAR motifs as the gold-standard for TF binding sites is not overly optimistic. An alternative would be to use the reproducible/rank consistent peaks based on the irreproducible discovery rate (from encodeproject.org) as the gold-standard peak set. In addition, a comparison of the results to existing methods for this task, such as PIQ (Sherwood et al., Nature Biotechnology, 2014) and Iterative CENTIPEDE (Moyerbrailean et al., PLoS Genetics, 2016), should be added.

Thank you for bringing these points.

Regarding the gold standard, we now took your advice and used the reproducible peaks (defined by ENCODE using irreproducible discovery rate (IDR) <0.02) overlapping with motifs as gold standard. The use of reproducible peaks directly addresses your concern about the quality of gold standard, and it does not change the main conclusions. The motif was still used in defining gold standard for two reasons. First, it is unrealistic to expect a prediction algorithm to infer identities of TFs binding to a DHS without motif information. This is a common fact for all TFBS prediction methods (including PIQ and CENTIPEDE) and it is not unique to BIRD. Because of this, our TFBS prediction was focused on analyzing DHSs that contain TFs' motifs. Consistently, motif-containing TF ChIP-seq peaks were used as gold standard. Second, ChIP-seq peaks without motifs may correspond to indirect TF-DNA association. The updated TFBS prediction results are shown in **Figure 3** and **Supplementary Figures 17-27**. The texts in the article are also updated accordingly.

Regarding comparison with existing methods, we compared BIRD with PIQ and CENTIPEDE. Both PIQ and CENTIPEDE use true DNase-seq data and motif information to make predictions. PIQ showed comparable performance with our true DNase-seq method, whereas CENTIPEDE performed worse than BIRD. Thus, replacing our true DNase-seq method by PIQ or CENTIPEDE as positive controls did not change the main conclusion regarding the usefulness of BIRD. The comparison of BIRD, PIQ and CENTIPEDE are presented in detail in **Supplementary Notes S3** and **Supplementary Figures. 20-23 and 26-27**.

We have also made serious efforts to run iterative CENTIPEDE, but there is no package or user-friendly software available. We can download the source code of iterative CENTIPEDE, but there is no manual or guideline explaining how to use it. We have contacted the authors multiple times but did not get their response. See emails below:

wzhou14@jhu.edu

Mon 7/18, 11:10 AM

rpique@wayne.edu ✕

  Reply all | 
Dear Dr. Pique-Regi,

This is Weiqiang Zhou.

I read your paper "Which Genetics Variants in DNase-Seq Footprints Are More Likely to Alter Binding?". It is very interesting.

I wonder if the iterative CENTIPEDE program is available as a package for the users? I am working on TF binding site prediction and would like to run iterative CENTIPEDE on my data.

Thank you.

Regards,

Weiqiang

wzhou14@jhu.edu

Thu 8/18, 1:48 PM

rpique@wayne.edu ✕

  Reply all | 
Dear Dr. Pique-Regi,

Sorry to bother you again.

This is Weiqiang Zhou.

I read your paper "Which Genetics Variants in DNase-Seq Footprints Are More Likely to Alter Binding?". It is very interesting.

I wonder if the iterative CENTIPEDE program is available as a package for the users? I am working on TF binding site prediction and would like to run iterative CENTIPEDE on my data.

Thank you.

Regards,

Weiqiang

-The authors show that their method is superior to chromimpute "on average". The most important application of their method would be to predict DH sites in previously poorly studied cell types, but in such cell types the primary goal is typically to identify DH sites that are differentially accessible when compared to a closely related cell type (eg neural progenitor differentiation). Unfortunately a method that appears to perform well on average may do poorly at predicting such changes. The authors should repeat their analysis when restricted to regions that are differentially expressed in closely related cell types.

This is a good point. We note that our ability to evaluate the performance of predicting differentially accessible regions is constrained by the availability of cell types and biological conditions that have both exon array (for making predictions) and regulome data (for evaluating the predictions). Currently, there are not many such datasets available. Therefore, in the revised manuscript, we have added three additional analyses to evaluate our method's ability to detect differentially accessible regions.

First, we evaluated our ability to predict differential DH both in all DHS regions and in differential DHS regions using the 17 ENCODE test cell types. This analysis is presented in the revised manuscript as follows: *"Since one potential application of BIRD is to predict differential DH between two sample*

types, we further evaluated BIRD in this regard by conducting pairwise comparisons of the 17 test cell types. Cell type pairs were stratified into four equal-sized groups based on the similarity level of the global DH profiles of two compared cell types. For each pair of cell types, the correlation between the predicted and true DH differences was computed across all loci and across differential loci respectively (**Methods**). For cell type pairs with the highest similarity level (i.e., those in the highest similarity quartile), the mean correlation was 0.42 for all loci and 0.55 for differential loci (**Fig. 2g**). For cell type pairs in the other similarity quartiles, the mean correlation was higher: 0.60-0.66 for all loci and 0.69-0.75 for differential loci (**Fig. 2g**). Again, compared to the prediction for individual loci, differential DH prediction at pathway-level was more accurate (**Fig. 2g**, mean correlation=0.60-0.77; **Supplementary Fig. 10**)."

Second, we have compared BIRD and ChromImpute in terms of their ability to predict differential DH using the 10 test cell types analyzed by both methods. The analysis was conducted in a way similar as above. This is presented in the revised manuscript as follows: "*We further applied both methods to predict differential DH between each pair of the 10 test cell types. BIRD outperformed ChromImpute substantially for comparing cell types with high similarity (i.e., cell type pairs whose similarity level was above the median), and ChromImpute performed slightly better than BIRD for comparing cell types with lower similarity levels (Fig. 2j, Supplementary Fig. 14e-f).*".

Third, we further tested the ability of BIRD to predict differentially accessible regions in a real setting. We applied BIRD to a new exon array dataset where human iPSC cells were differentiated into dopaminergic neurons, a cell type not contained in our training data. These exon array data are not generated by ENCODE. For these exon array samples, no corresponding experimental regulome data are available. This represents a typical scenario for which BIRD can be useful. Using BIRD, we predicted differential DH between samples before and after differentiation. We randomly sampled 26 DHSs at different predicted fold change levels along with 5 control DHSs, and we performed ChIP-qPCR validation on these 31 DHSs using an independent histone mark H3K4me1. ChIP-qPCR confirmed differential regulatory element activities for 15/22 (68%) DHSs with predicted $|\log_2 \text{fold change}| > 1$. According to our experience with other systems, this represents a very good empirical validation rate. At the $|\log_2 \text{fold change}| > 1$ cutoff, we obtained 76,495 differential DHSs, representing a significant amount of new information one would not have if BIRD were not available. Further analyses show that the differential DHSs predicted by BIRD were enriched in flanking regions of differentially expressed genes. We also identified enriched DNA motifs in predicted differential DHSs. For DHSs down-regulated in iPSC-derived neurons, enriched motifs contained TFs closely involved in stem cell maintenance such as SOX2, OCT4 (aka POU5F1), KLF4, and NANOG. For DHSs up-regulated in iPSC-derived neurons, enriched motifs contained TFs involved in neuronal differentiation. These include important regulators of neurogenesis and neural development such as NEUROG2, NEUROD2 and ZBTB18 (aka RP58), as well as ATOH1, a TF recently reported to play an important role in the differentiation of dopaminergic neurons. These analyses demonstrate how BIRD can be used in practice to predict differential regulatory signals when experimental regulome data are not available. The predicted differential DHSs and their potential regulators suggested by the enriched motifs may be used as candidates to guide follow-up functional experiments (e.g., knock-out experiments) to accelerate the study of regulatory circuitry. These results are presented in detail in the newly added results section "**Predicting Differential Regulatory Signals in a Differentiation System**", **Figure 6, Supplementary Figures 31-33 and Supplementary Tables 8-10**.

We note that in addition to the above analyses, our examples in **Figure 5** also demonstrate how BIRD can be used in practice to predict differential regulatory signals. Collectively, these analyses and results demonstrate that BIRD is capable of predicting differential accessibility.

- The Results section does not include any interpretation of the model. It would be great if a paragraph could be added on a few examples of specific genes or groups of genes whose expressions are predictive of specific accessibility regions. For example, I would be interested to see if promoters' accessibilities are often predicted mostly by their corresponding gene, if enhancers with known targets' accessibilities are often predicted mostly by those targets, or highly-conserved regions' accessibilities are often predicted by genes involved in pathways that are conserved across species.

This is a very good point. In the revised manuscript, we performed additional analyses to help interpret the models. We found that promoter DHSs' associated genes (i.e. the closest gene of each promoter DHS) and enhancer DHSs' target genes are enriched in their predictors compared to random expectation. However, the majority of promoter and enhancer DHSs do not have their closest or target genes chosen by BIRD as predictors. Note that since we used clusters of co-expressed genes as predictors, having (or not having) a gene in the predictor implies that its co-expressed genes are also included (or not included) in the predictors. For DHSs with phylogenetically conserved DNA sequences, we asked whether their predictors were more conserved. However, no clear association was found. We also found that TFs that potentially regulate DHS pathways are enriched in pathways' predictors.

These new analyses and results are presented in the new section "**Predictors selected by BIRD**" as follows:

*"BIRD predicts the DH level of a DHS by aggregating locus-level and pathway-level predictions via a weighted average. Therefore, predictors of each DHS consist of both predictors from the locus-level model and predictors from the pathway-level models. Analyses of the predictors selected by BIRD show that only 5.2% of all DHSs, 9.8% of promoter DHSs and 13.6% of enhancer DHSs had their closest genes or target genes contained in the predictors. While these represent significant enrichment (7-fold, 13-fold and 7-fold enrichment respectively, permutation test p -values < 0.001 , **Methods, Supplementary Fig. 15a-c**) compared to random expectation, the majority of DHSs did not have their closest genes or target genes chosen as predictors, consistent with the hypothesis that information useful for prediction is not all contained in DHSs' closest or target genes. For DHSs with phylogenetically conserved DNA sequences, we asked whether their predictors were more conserved than typical. However, no clear association was found (**Supplementary Notes, Supplementary Fig. 16**).*

*We further analyzed predictors selected by the pathway-level models (i.e., $BIR(\bar{X}, \bar{Y})$) for each DHS pathway. It was found that TFs that potentially regulate the pathways were enriched in pathways' predictors. For example, consider clustering DHSs into 1000 pathways. We identified enriched DNA motifs for each DHS pathway. For 37.8% of the DHS pathways, at least one TF corresponding to the enriched motifs was contained in the predictors selected by $BIR(\bar{X}, \bar{Y})$. By contrast, if the same number of predictors were randomly assigned to each DHS pathway, only 9.9% of the DHS pathways would have their predictors covering at least one TF for the enriched motifs (37.8%/9.9% = 3.8-fold enrichment). Clustering DHSs into 2000 or 5000 pathways yielded similar results (**Supplementary Fig. 15d-f**). Since we used clusters of co-expressed genes as predictors, having a TF in the predictor implies that genes co-expressed with the TF, which often come from related biological processes or pathways, are also included in the predictors. **Supplementary Figure 15g-j** and **Supplementary Table 2** show a sample DHS pathway which is active in human skeletal muscle myoblasts (HSMM) and skeletal muscle myotubes differentiated from HSMM (HSMMtube) (**Methods**). TFs known to be involved in muscle development such as MYF6, MYOD1 and MYOG were found in the pathway's predictor genes (**Supplementary Figure 15i**), and motifs of these TFs were among the top enriched motifs in the DNA sequences of DHSs in this pathway (**Supplementary Figure 15h**). Gene Ontology (GO) analysis of the predictor genes further identified "muscle cell development" and "skeletal muscle*

tissue development” as enriched GO terms for this pathway (Supplementary Figure 15j). Thus, predictor genes of this pathway were enriched in biological processes consistent with the cell types in which the DHS pathway is active. The same analyses were also conducted on other pathways.”

The complete catalog of BIRD-selected predictor genes for each DHS and DHS pathway, and the enriched DNA motifs and GO terms for each pathway are provided at <https://zhiji.shinyapps.io/CABS/> as an online resource.

Appropriate use of statistics and treatment of uncertainties:

The analysis is generally well presented and accurate, with a couple exceptions listed below.

-The authors (mostly) use Pearson correlation as their similarity measure, but they should include a ranked based measure as well.

This is a good point. We actually tried Spearman’s rank correlation too. It did not change the conclusions of this study. To demonstrate, the newly added **Supplementary Figs. 12-13** compared different methods using Spearman’s correlation. One can see that all main conclusions remained the same. For example, BIRD still outperformed the neighboring gene approach. Clustering correlated predictors ($BIR(\bar{X}, Y)$) still performed better than not clustering predictors ($BIR(X, Y)$). The cross-locus prediction accuracy r_L remained higher than the cross-cell-type prediction accuracy r_C . Pathway-level prediction remained more accurate than locus-level prediction in terms of cross-cell-type accuracy.

In the revised manuscript, we discussed this point:

“Our previous analyses used Pearson’s correlation as the correlation measure. The conclusions remained qualitatively the same when the Spearman’s rank correlation was used instead (Methods, Supplementary Fig. 12-13).”

-All of the sensitivity/predicted site graphs should be converted into the more standard sensitivity/specificity plot

We have added ROC curves for TFBS prediction. Please see **Figure 3b,e** and **Supplementary Figures 18, 21, 23b, 24d-f, 25b, 26d-f, 27b**.

We also provided sensitivity versus FDR curve. This is because ROC which plots sensitivity versus 1-specificity is known to be misleading in cases where the negative cases far-outnumbers the positive cases. For example, suppose there are 10 true binding sites among a total of 10000 genomic sites. Suppose one wants to predict binding sites, and the 10 true binding sites are distributed uniformly among the top 100 predictions (i.e., the predictions ranked 1-10 contains one true binding site, the predictions ranked 11-20 contains one true binding site, etc.). Then for these top 100 predictions, the sensitivity is 100%, and the specificity is $(9990-90)/9990=99\%$. Both the sensitivity and specificity are high. This may make people think that the prediction was great. However, this is not the case because among the top 100 predictions, 90% are false discoveries (i.e. $FDR=90\%$). The problem with ROC is that it does not reveal this issue. For this reason, ROC does not provide a picture that can be used to directly tell the practical value of a prediction method. For this reason, we plotted sensitivity versus FDR in the revised manuscript to more directly illustrate the performance of different prediction methods which are more relevant to their practical use.

Conclusions:

This method has potential to be very useful, but the results are a bit weak if the method is primarily being sold as a predictive model. As written, it confirms the expected results that gene expression is on average predictive of accessibility, and shows that clustering predictors can improve prediction accuracy. These results are interesting but the ability to predict the cell-type specific DHSs (which are arguably the most relevant and hardest to predict) accurately is not clearly demonstrated.

We appreciate the reviewer's comments. We would like to respond from multiple perspectives.

First, taking the reviewers' advice, we have added a number of new analyses in the revised manuscript which have greatly strengthened our results. In particular, in order to test BIRD's ability to predict cell-type specific DHSs, we added a new example in which BIRD was applied to a new exon array dataset where human iPS cells were differentiated into dopaminergic neurons, a new cell type not contained in the training data. For these cells, corresponding DNase-seq data were not available. For BIRD predictions randomly selected at different rank levels, we performed ChIP-qPCR validation using an independent histone mark H3K4me1. These results demonstrate the ability of BIRD to predict differential DH activities in different sample types in practice. We also expanded our analysis of the 17 ENCODE cell types to include the analysis of BIRD predictions of pairwise differences between two cell types.

Second, we wish to point out that while prediction method is an important component of this study, our study is more than just presenting a predictive model. It represents the first systematic analysis of the regulome and transcriptome relationship in its reverse direction. Besides presenting a prediction method BIRD, we also illustrate for the first time several new ways to use gene expression data to help tackle several challenging problems in the study of gene regulation, such as TFBSs prediction when experimental regulome data are not available, regulome mapping in massive number of sample types, and integrating different data types, etc.

Third, our results are more than just confirming "the expected results". Note that before us, no one has systematically tested the idea of using gene expression alone to predict chromatin accessibility. As such, no one knows whether using gene expression alone can predict DHS is feasible, to what extent the chromatin accessibility can be predicted using gene expression, and whether the computation can be done efficiently and how. Our study is the first of this kind trying to provide a systematic answer to these questions. Looking at our results retrospectively, everything seems to make sense. However, readers tend to ignore the fact that these answers were not obvious at all before one conducts this study. For example, without such a systematic analysis, we did not know the answer regarding whether neighboring genes are sufficient for prediction and how much improvement using the whole transcriptome can bring. For this reason, this study is important in that it provides the first comprehensive picture of how gene expression may predict chromatin accessibility.

Forth, in terms of prediction method, constructing a method that can efficiently handle the data where both predictors and responses are high-dimensional is non-trivial (at least not as trivial and "expected" as this reviewer thought). Applying existing methods do not always work for these noisy real data with high-dimensionality. For example, we compared BIRD with several popular group variable selection methods as suggested by the reviewer. However, those methods are mainly designed for univariate response instead of high-dimensional responses. Systematic evaluation of those methods in real applications with high-dimensional responses and predictors is still lacking. In fact, these methods run substantially slower than BIRD and often had lower prediction accuracy. None of these methods tell people that clustering correlated predictors and using clustering in the way as in BIRD provide the most effective and efficient solution to handling the kind of data we have (which is noisy and had

high-dimension in both predictors and responses). BIRD is a method we developed after tremendous amounts of trial-and-error work. It is tested using noisy real data with high-dimensionality in both responses and predictors. Lessons learnt in this study therefore can greatly benefit the whole scientific community since they could help researchers facing similar big data problems to more quickly develop a good solution in the future. We learned a lot from this study ourselves. We are confident that reporting our methods and findings here will be tremendously useful for others too.

Finally, our results for the first time demonstrate the feasibility of using gene expression data to predict TFBSs when experimental regulome data are not available, to map regulome in massive number of sample types, and to serve as pseudo-replicates for data integration. These new ways to use gene expression data have not been explored before, and none of these results can be taken for granted unless one really conducts the study to see what happens. Given the wide use of gene expression data, our findings could have great impact on future genomic studies. Our predictions may not be perfect, but it is important to understand that the prediction performance presented here may not have reached its upper limit. For example, for the TFBS prediction, it is very likely that future research could further improve our prediction accuracy by improving DNA motif models and further incorporating other types of information (e.g., phylogenetic conservation, locations relative to genes, collaborating motifs). Obviously, it is impossible to explore everything within a single research article, nor is it realistic to expect that one can provide the optimal solution to all these applications which are all quite complex and have not been studied before. Nonetheless, this does not mean that our results are weak or not significant. To understand this, one can just look at two historical examples. First, the ENCODE pilot project only studied 1% of the human genome. Despite its low coverage of the genome, it is equally important as the expanded ENCODE project that studies the whole genome. This is because the pilot project establishes the feasibility which provides the basis that later research can build upon. Second, for many important prediction problems such as speech recognition, the original prediction accuracy was not high and far from enough for practical use. However, through decades of continual research, the prediction accuracy has been improved dramatically to a level that predictions can now be applied in many real-life settings. Although the initial introduction and feasibility study of the concept that computer may be used to translate spoken language to texts was accompanied with relatively low prediction accuracy, it still represents the most important step in that field because it opened the door of a new research direction. Similarly, the significance of our study cannot be evaluated only based on how accurate we can predict now. A major contribution of our study is the new opportunities we show that one can have with transcriptome-based prediction of regulome, and our demonstration of the feasibility in these previously unexplored areas.

Suggested improvements:

This manuscript would be much more interesting if the authors were able to show that their method could be used to predict differentially accessible regions. The authors' method also needs to be compared to other methods from the group selection literature.

Thank you for these thoughtful suggestions. In the revised manuscript, we have added new analyses to demonstrate that BIRD can be used to predict differentially accessible regions. We have also compared BIRD with other group selection methods including group lasso, composite MCP, and fused lasso. Our detailed responses to both these comments have been provided above and will not be repeated here.

References:

Again, the authors need to discuss the major group selection methods.

We have added a discussion about the connection between BIRD and group selection methods, and we also compared BIRD with group lasso, composite MCP, and fussed lasso (see our responses above).

Clarity and context:

The abstract and conclusion are accurate and clear.

Thanks for these comments!

Reviewer #2 (Remarks to the Author):

To the authors:

Summary: The authors develop an approach for predicting DNASE hyper sensitive (DH) sites using gene expression data. This approach is novel in that it's the first attempt to analyze these data in a reverse fashion: mRNA \rightarrow DH instead of vice versa. The author's model BIRD has several variations. In addition, the BIRD approach has carefully taken into account factors related to the computational and statistical challenges associated with processing massive datasets. They have applied their approach to multiple data scenarios and have compared with existing methods, permuted data, and compared its potential improvement in enhancing sparse or low-quality data (so called pseudo replicates).

Thank you for the careful reading and summary of our manuscript.

Major comments:

1. Overall, the authors present several BIRD models. The flow/presentation of these models is in a somewhat linear fashion and is very confusing. It is still unclear to the reviewer what are the distinct advantages of each (are they all needed?). Can't one or two of these approaches be used instead of all of them?

Thank you for raising this point. We are sorry for the confusion. In fact, there is only one final BIRD model, which is the one that aggregates the locus-level model $BIR(\bar{X}, Y)$ and the pathway-level model $BIR(\bar{X}, \bar{Y})$. The other models including $BIR(\bar{X}, Y)$, $BIR(X, Y)$, $BIR(\bar{X}, \bar{Y})$ are all intermediate models used to help us and readers understand how different data analysis techniques affect prediction performance. More specifically, the comparison between $BIR(\bar{X}, Y)$ and $BIR(X, Y)$ shows that clustering correlated predictors are useful. The comparison between $BIR(\bar{X}, Y)$ and $BIR(\bar{X}, \bar{Y})$ shows that predicting the DHS pathway's overall activity is easier than predicting the activity of individual locus. The comparison between BIRD and $BIR(\bar{X}, Y)$ shows that aggregating locus-level model with pathway-level model further improves performance. In other words, these intermediate models are used as a tool to evaluate effectiveness of various model building techniques, which we hope can help readers to understand the rationale behind the final BIRD model.

The reason why we wanted to show these intermediate models is because the research on big data regression and prediction problems with high-dimensionality in both predictors and responses is still at its early stage. In this field, answers to important questions such as which techniques are useful and provide both high accuracy and computational efficiency are still lacking and incomplete. While biologists may be only interested in how accurate we can predict DNase-seq from gene expression, data scientists, statisticians and computer scientists are also interested in what lessons we can learn

from this genomic application in terms of general methodology for big data analysis. Such lessons could potentially be useful for many other big data problems such as those in neuroimaging, health data, etc. Our study provides a valuable opportunity to test the effectiveness of different model building techniques in noisy real data. In this regard, lessons learned from comparing the intermediate models above can shed light on what techniques might be useful for similar big data problems in the future. For this reason, although there is only one single final BIRD model (which is used in all sections to predict TFBSs, construct PDDB, and predict differential DH signals), we think it is still useful to compare these intermediate models.

We do appreciate your feedback that presenting all these intermediate models may cause some confusion. In order to address this, in the current manuscript, we revised the structure of presentation as follows:

- 1) When we introduce BIRD, we first make it clear that:

“BIRD predicts DH at each genomic locus by combining predictions from two types of models, locus-level model and pathway-level model, through model aggregation (Fig. 1b).”

After making this statement, we then proceed to describe the locus-level and pathway-level model. Next, we make it explicit again that the final BIRD prediction is obtained by aggregating these intermediate models:

“The final prediction by BIRD for each genomic locus is a weighted average of the locus-level and pathway-level predictions. The intuition is that the pathway activity predicted by $BIR(\bar{X}, \bar{Y})$ can also serve as a prediction of the DH level for each individual locus within the pathway. Such a prediction may be biased but less noisy than the locus-level prediction by $BIR(\bar{X}, Y)$. Thus, integrating the locus-level prediction from $BIR(\bar{X}, Y)$ and the pathway-level prediction from $BIR(\bar{X}, \bar{Y})$ through model aggregation may result in a better tradeoff between the prediction bias and variance. In fact, the aggregated model robustly improved the overall locus-level prediction accuracy (Supplementary Fig. 4, Methods). Based on these results, we used the aggregated model, termed BIRD, in all subsequent analyses to predict DH levels for each individual locus.”

- 2) The systematic performance comparisons of the intermediate models (which were used to reveal the effectiveness of different techniques) are now moved to supplementary materials. Major findings from these comparisons are briefly summarized in the main article. The revised main manuscript is now primarily focused on discussing the final BIRD model.

We hope that this can help avoid the confusion mentioned by the reviewer.

2. It seems that overall the improvements of BIRD is fairly low. For example, the MYC example was very low (~56%). The authors need to do a better job justifying why these results are useful in practice by better justifying the benefits of their predictions

Thank you for raising this point. In the original manuscript, we may not have made it clear, but the improvements of BIRD over existing methods are actually substantial. As another reviewer commented, MYC is not a good example to demonstrate this because E-box motif can be bound by too many other TFs, making the accurate estimation of false discovery rate and method evaluation difficult. Taking that reviewer’s suggestion, we have changed the example from MYC to ELF1 to illustrate the TFBS prediction. Using this new example and a number of new analyses in the revised manuscript, below we address your question from multiple perspectives.

(1) Does BIRD bring substantial improvements?

We would like to point out that the amount of improvements by BIRD depends on which method is used as the baseline to compare with. Below we discuss several baselines.

Motif-only baseline: When experimental regulome data (e.g. DNase-seq) are not available, currently the standard method to predict transcription factor binding sites is based on DNA motifs. This motif-only baseline was not compared in our old manuscript. However, it is the most commonly used method in practice when no experimental regulome data are available. Note that the other baseline method used by us which predicts TFBSs by coupling mean DH in the training data with motif information did not exist before our study. Therefore, in order to provide a fair answer to the question of how much we have improved over the current practice, the motif-only baseline is a more appropriate baseline to use. We now added the motif-only baseline to the revised manuscript. Compared to TFBS predictions based on DNA motif only, the gain brought by BIRD is substantial. For example, **Figure 3a** shows the sensitivity of predicting ELF1 binding sites at different FDR levels for different methods. At 10%, 25%, and 50% FDR level, BIRD predictions gave a sensitivity of 0.43, 0.64, and 0.88 respectively, as compared to 0, 0.02, and 0.11 by the motif-only approach. At these FDR levels, BIRD generated 5000, 9000, and 19000 predicted ELF1 binding sites, which were substantially more than the number of the predicted sites by the motif only (0, 250 and 2000) (**Fig. 3c**). **Figure 3d-e** compares the area under the sensitivity-FDR curve and the area under the ROC curve (AUSFC and AUROC) of different methods for different TFs. BIRD was substantially better than the motif-only approach. Take ELF1 as an example, BIRD had AUSFC=0.61 and AUROC=0.93. As a comparison, the motif-only approach had AUSFC=0.16 and AUROC=0.65.

Mean DH baseline: The mean DH level in the training cell types coupled with motif information predicts TFBSs much better than the motif-only method. This is why the mean DH approach was presented in our old manuscript as the baseline. There, we did not present the motif-only baseline. This may have sent a misleading message that mean DH is the state-of-the-art method used in practice. But actually it is not. No previous study has attempted to couple mean DH profile with DNA motifs to predict TFBSs. Thus, predicting TFBSs using mean DH is a new method itself. It has not been used or tested before. That said, BIRD still brought substantial improvement over the mean DH approach (**Figure 3**). For instance, in the ELF1 analysis, the sensitivity of BIRD predictions at 10%, 25%, and 50% FDR level was 0.43, 0.64, and 0.88 respectively, as compared to 0.09, 0.36, and 0.62 by the mean DH approach (**Fig. 3a**). At these FDR levels, BIRD generated 5000, 9000, and 19000 predicted ELF1 binding sites, which is substantially more than 1000, 5000, and 13000 predicted binding sites by the mean DH approach (**Fig. 3c**). The area under the sensitivity-FDR curve and ROC curve (AUSFC for BIRD (AUSFC=0.61, AUROC=0.93) were also substantially better than the mean DH approach (AUSFC=0.43, AUROC=0.82) (**Figure 3d-e**).

True DNase-seq profile (true DH): The true DH approach is used as a positive control. It is the performance one would achieve when experimental DNase-seq data are available. One would expect it to perform better than BIRD. Compared to the true DH approach, BIRD still performed okay (**Figure 3**). However, BIRD is primarily used when there is no experimental DNase-seq data. In those situations, the true DH approach obviously is not applicable, and one would not get any prediction (sensitivity = 0) using the true DH approach.

Collectively, this discussion shows that the improvements of BIRD for TFBS prediction are substantial.

(2) Are the predictions useful when there are prediction errors?

As a prediction method, prediction errors are inevitable. Although TFBS predictions by BIRD are not perfect, they can still be very useful. Suppose one finds a set of interesting genes from a gene expression experiments and wants to identify key TFBSs of these genes in the biological context under

study, and suppose DNase-seq or other regulome data are not immediately available due to technical or resource constraints. In this situation, one would face two options. In option 1, one waits many years and does nothing until the experimental regulome data become available. In option 2, instead of waiting, one uses BIRD to predict regulatory elements active in the biological context in question. Although some predictions may be incorrect, the majority are still correct as long as $FDR < 50\%$. One can then use the predictions to narrow down the list of candidate TFBSs and pick up the most promising candidates to follow-up. Using this approach, one can learn something now without waiting. Which option would one choose? This is a common situation faced by many laboratories and investigators. If we face this situation, we would choose option 2. Moreover, FDR is adjustable. If one wants a smaller FDR, one can choose a more stringent cutoff to reduce FDR at the cost of losing some sensitivity. For example, in the ELF1 example, the sensitivity decreases from 0.88 to 0.64 when the FDR is reduced from 50% to 25%. Still, one gets 9000 predicted ELF1 binding sites at the 25% FDR level. That is a lot compared to knowing nothing if one chooses to wait instead of using BIRD to make prediction. Thus, even if BIRD prediction may not be perfect and may not be as good as true DNase-seq-based TFBS prediction, it can still provide tremendous amounts of useful information to guide follow-up studies.

(3) How to interpret the prediction performance presented in this study?

It is important to keep in mind that the TFBS prediction performance presented in this article does not represent the upper limit one can achieve using BIRD. TFBS prediction is a complex problem involving integration of multiple types of information such as chromatin accessibility and DNA motif, etc. Our analyses here were focused on investigating different ways to use one information type (i.e., chromatin accessibility). There is still plenty of room for improving TFBS prediction accuracy by optimizing the use of other information types. For instance, by using a more sophisticated motif model that accounts for the intra-motif correlation (i.e., correlation among different positions within a motif), we could increase the sensitivity of BIRD prediction for ELF1 at the 10%, 25% and 50% FDR level from 0.43, 0.64 and 0.88 to 0.51, 0.75 and 0.88. The sensitivity at 10% and 25% FDR level outperformed the prediction based on true DNase-seq coupled with the conventional motif model which ignores intra-motif correlation (**Supplementary Notes, Supplementary Fig. 28**). Besides intra-motif correlation, information such as phylogenetic conservation, motifs of collaborating TFs, and genomic locations relative to genes may also improve TFBS prediction. However, the focus of this article is to illustrate the use of predicted DH in the absence of experimental regulome data rather than building the optimal TFBS prediction pipeline. As such, systematically exploring the optimal use of all these non-DH predictors in TFBS prediction is beyond the scope of this study. For this reason, the significance of our results cannot be evaluated just based on the presented prediction accuracy. The most significant and useful thing we contributed through our TFBS analysis is our demonstration of the feasibility of using transcriptome to predict TFBSs when experimental regulome data are not available. This adds a new tool to the toolbox one can use to build better TFBS prediction pipelines in the future.

(4) Others

TFBS prediction is not the only application where BIRD is useful. In fact we have shown that BIRD is also useful in many other applications, such as expanding the regulome catalog by using gene expression samples in GEO, or providing a bridge to integrate different data types. Also, in response to your question 4, we added a new example in the revised manuscript to further demonstrate the practical use of BIRD in predicting differential regulatory element activities in a differentiation system with good accuracy (see responses to question (4) below and the new results section "**Predicting Differential Regulatory Signals in a Differentiation System**"). Collectively, these examples provide substantial evidence to show that having BIRD is useful and beneficial in practice.

3. Clustering: K means were used and compared with unclustered variable selection. How about basing the BIRD regression models using other dimension reduction approaches such as PCA? (e.g. use PCA components in the regression instead of K means).

Thank you for making a very good point. In the revised manuscript, we have compared our method with principal component regression. Using principal components instead of K means to reduce the dimension actually performed worse (see **Supplementary Notes S1** and **Supplementary Fig. 5**).

4. One important analysis that seems lacking would be in BIRD's ability to identify differential DH regions from differentially expressed genes in a treatment/control or disease/no disease state. Would be very useful to see if BIRD works in this case.

This is a very good point. In the revised manuscript, we have applied BIRD to predict differentially accessible regions. More specifically, we applied BIRD to a new exon array dataset where human iPSC cells were differentiated into dopaminergic neurons, a new cell type not contained in the training data. For these exon array samples, no corresponding experimental regulome data are available. This represents a typical scenario for which BIRD can be useful. Using BIRD, we predicted differential DH between samples before and after differentiation. For BIRD predictions randomly selected at different rank levels, we performed ChIP-qPCR validation using an independent histone mark H3K4me1. The results show that BIRD was able to predict differential regulatory element activity with good accuracy (15/22 (68%) tested DHSs predicted to be differential were validated by H3K4me1 ChIP-qPCR). In total, BIRD generated 76,495 differential DHSs, representing a significant amount of new information one would not have if BIRD were not available.

Further analyses show that the differential DHSs predicted by BIRD were enriched in flanking regions of differentially expressed genes. DNA motifs enriched in predicted differential DHSs were also found to be consistent with the biology of this system. For DHSs down-regulated in iPSC-derived neurons, enriched motifs contained TFs closely involved in stem cell maintenance such as SOX2, OCT4 (aka POU5F1), KLF4, and NANOG. For DHSs up-regulated in iPSC-derived neurons, enriched motifs contained TFs involved in neuronal differentiation. These include important regulators of neurogenesis and neural development such as NEUROG2, NEUROD2 and ZBTB18 (aka RP58), as well as ATOH1, a TF recently reported to play an important role in the differentiation of dopaminergic neurons. These analyses demonstrate how BIRD can be used in practice to predict differential regulatory signals when experimental regulome data are not available. The predicted differential DHSs and their potential regulators suggested by the enriched motifs may be used as candidates to guide follow-up functional experiments (e.g., knock-out experiments) to accelerate the study of regulatory circuitry. These results are presented in detail in the newly added results section "**Predicting Differential Regulatory Signals in a Differentiation System**", **Figure 6, Supplementary Figures 31-33 and Supplementary Tables 8-10.**

Reviewer #3 (Remarks to the Author):

The paper presents BIRD, a method to predict the DNase I hypersensitivity signal of predefined genomic locations (which were detected as DHS in at least one of the training samples) based on the transcriptional activity in the whole genome. This means that it cannot discover new (previously undetected) DHS, only predict the hypersensitivity of those that are already known to be hypersensitive in some cell lines/samples. It compares the performance of several variants of the method with a simpler approach, as well as with a related existing software (ChromInpute, which is more general but requires multiple types of experiments as input, most commonly ChIP-seq for multiple histone modifications). It also claims to provide the results of the method applied to 2000 existing exon array samples as a web resource (PDDB).

The paper is reasonably well-written and the points made in the text generally well supported by figures, and the methods are clearly explained. I am yet to be convinced in its broad practical utility. Since its predictions are based on training on existing data, no new cell-type specific biology is likely to be inferred, even though the predictions based on what is already known can be useful.

Thank you for your careful reading and summary of our manuscript. We appreciate your understanding that “predictions based on what is already known can be useful.”

Regarding broad practical utility, we added several new analyses based on reviewers’ comments to strengthen our results. In our old manuscript, we have already demonstrated the application of BIRD in TFBS prediction, converting GEO into a regulome database, and serving as a bridge to integrate different data types. In the revised manuscript, we further strengthened the results in TFBS prediction, added a new application in which BIRD is used to predict differential DH in a realistic setting, and showed the potential of using BIRD for assessing data quality. For example, we have applied BIRD to a new exon array dataset where human iPS cells were differentiated into dopaminergic neurons, a new cell type not contained in the training data. For these cells, the corresponding DNase-seq data were not available. We randomly selected BIRD predictions at different rank levels and performed ChIP-qPCR validation using an independent histone mark H3K4me1. The results show that BIRD was able to predict differential regulatory element activities in this differentiation system with good accuracy (see new results section “**Predicting Differential Regulatory Signals in a Differentiation System**”). In studies like this, BIRD predictions can provide candidate regulatory elements to design follow-up experiments to guide mechanism studies of genes’ transcriptional control. In addition to these new analyses and examples, we would also like to highlight the following bioRxiv preprint where we demonstrate the utility of BIRD in RNA-seq, including RNA-seq from samples with small number of cells and single-cell RNA-seq (Zhou, W., Ji, Z. & Ji, H. *Global Prediction of Chromatin Accessibility Using RNA-seq from Small Number of Cells*. *bioRxiv*, 035816. (<http://www.biorxiv.org/content/early/2016/01/01/035816>)). There we show that BIRD prediction in small-cell-number samples can outperform or improve direct experimental measurements by ATAC-seq and single-cell ATAC-seq. Together, these results demonstrate the broad practical utility of BIRD.

Regarding prediction in new cell types, we note that for mapping regulatory element activities, there are two different issues to address: (1) finding locations of all regulatory elements in a genome and (2) documenting activities of each element in all biological contexts (i.e. different cell types and conditions). Both issues are important. Ideally, one would like to address both. Realistically, however, no existing method can effectively address both issues. The chromatin accessibility experiments are very good at locating regulatory elements in a new sample, but they do not offer the efficiency required for mapping regulatory element activities in a massive number of samples and biological contexts. Our BIRD approach, on the other hand, assumes that locations of regulatory elements in the genome are given. As such, it is not designed to find new locations of regulatory elements that never

occur in the training data. However, given regulatory element locations, BIRD allows one to predict regulatory element activities in a massive number of samples and biological contexts which cannot be efficiently done by chromatin accessibility experiments. Thus, the combination of chromatin accessibility experiments and BIRD prediction can provide a realistic solution to mapping both the locations and activities of cis-regulatory elements.

In terms of predicting activities of known regulatory elements in new cell types, our evaluation of BIRD was based on using test cell types not contained in the training cell types. Thus, our results have demonstrated the ability of BIRD to predict activities of regulatory elements in new cell types. Like any other prediction method, we do not expect BIRD prediction to be perfect. However, prediction can help researchers in multiple ways to study gene regulation. Examples include but are not limited to identifying promising regulatory elements active in a sample type of interest for follow-up studies (e.g., to guide knock-out experiments), developing hypotheses regarding how a gene is controlled, and using prediction as a bridge to integrate multiple data types. As such, BIRD predictions are tremendously useful even if they are not perfect.

In terms of finding new locations of regulatory elements in new cell types, it is not surprising that BIRD cannot discover new locations of regulatory elements in a new cell type because a supervised prediction approach such as BIRD simply has no information to build a prediction model for a location without any DNase-seq signal in any training sample. Fortunately, DNase-seq and other experimental regulome profile data are steadily growing. Even though experimental regulome data in all possible biological contexts will not be available in the near future, efforts such as ENCODE and Roadmap Epigenomics will soon provide regulome profiles along with gene expression data in a number of representative cell types and conditions. Using these data, it is very likely that most genomic loci in the human genome that can serve as regulatory elements will be discovered. In fact, analysis of the relationship between the number of regulatory elements and number of training cell types in **Supplementary Figure 39** (also shown below) indicates that the increase in the number of DHSs one can discover gradually slows down as the number of training cell type increases (see **Supplementary Notes S6** for details). It also shows that the undiscovered new regulatory elements in any new sample are likely to account for only a small fraction of the genome when a few dozens of training cell types are available. As the training data expands in the next couple of years, BIRD can be easily re-trained using the expanded training sets which likely will cover the vast majority of regulatory elements in the genome. It is important to keep in mind that even though locations of most regulatory elements may be discovered in the near future using the available regulome data, activities of these elements in the vast majority of biological contexts remain unknown and BIRD will help fill this vacuum. Thus, BIRD has its unique value which can greatly benefit the research community. These are now discussed in the **Discussion** section and **Supplementary Notes**.

Supplementary Figure 39. The impact on the discovery of new DHSs by adding a new cell type to training data. (a) The number of DHSs (denoted by $Y(n)$) discovered from training data is shown as a function of the number of training cell types (denoted by n). Mean and standard deviation from 10 simulations are shown (see **Supplementary Notes**) (b) Percentage of new DHSs contributed uniquely by adding a new cell type. This plot shows $[Y(n)-Y(n-1)]/Y(n)$ as a function of n . The mean from 10 simulations is shown.

Specific points:

Results, 2nd section, paragraph 3, states "transcription of a gene may be controlled by multiple cis-regulatory elements, DH of a particular regulatory element may not always correlate well with neighboring gene expression". This is not just because of multiple elements, but the fact that many elements are known to control genes over a very long genomic distance, in many cases across other genes.

This is a good point. We have updated the manuscript to include this additional possibility in our discussion:

"Biologically this is reasonable since many regulatory elements are known to control genes over a long genomic distance, sometimes across many other genes. Also, DH of a locus may be correlated in trans with expression of TFs that bind to the locus, genes that co-express with these TFs, and genes that co-express with the target gene controlled in cis by the locus. Moreover, since cell-type-specific transcription of a gene may be controlled by multiple cis-regulatory elements, a gene's expression

may not always correlate well with the DH level of each individual regulatory element in its neighborhood “

The first two test statistics used in Figure 1 could be better explained, with reference to the underlying biology, as "Cross-locus correlation" and "cross cell-type correlation" are not easy to interpret at first glance, and they are important to the section, e.g. "Relative levels of DH within a cell type can be accurately predicted", and "Variation of a single DH site across cell types..." The authors do give a good explanation of the causes of the discrepancy in performance, which is convincing.

Thank you for this nice suggestion. In order to help readers better understand the meaning of "Cross-locus correlation" and "cross cell-type correlation", we added the following explanation when these two concepts are first introduced in the manuscript:

*“For each locus, prediction models were constructed using the 40 training cell types. Prediction performance was evaluated using the 17 test cell types based on three types of statistics (**Fig. 1c, Methods**). First, we evaluated a method’s ability to predict the variation of DH levels across different genomic loci by computing the Pearson correlation between the predicted and true DH values (or P-T correlation) across all DHSs within the same cell type (cross-locus correlation r_L). Second, we evaluated a method’s ability to predict the relative activities of the same DHS in different cell types by computing the P-T correlation across different cell types at each genomic locus (cross-cell-type correlation r_C). Third, we computed the total squared prediction error normalized by the total DH data variance (τ) to characterize the proportion of data variation not explained by the prediction.”*

There is a brief mention of "pathway" in the description of the BIRD \bar{x} \bar{y} model, however it would be interesting if a further analysis of the clusters was given. Naïvely, it could be expected that each cluster in X could represent a transcription factor and its downstream targets (or multiple, closely related TFs), and Y might represent the binding sites for that TF and any downstream TFs. This could represent one of the most interesting findings in the paper.

This is an excellent point. Following your advice, we have conducted further analysis of the DHS clusters (i.e., DHS pathways) and gene clusters selected to predict each DHS cluster. The analysis shows that TFs that potentially regulate DHS pathways were enriched in pathways’ predictors. For example, consider clustering DHSs into 1000 pathways. We identified enriched DNA motifs for each DHS pathway. For 37.8% of the DHS pathways, at least one TF corresponding to the enriched motifs was contained in the predictors selected by BIR(\bar{X} , \bar{Y}). By contrast, if the same number of predictors were randomly assigned to each DHS pathway, only 9.9% of the DHS pathways would have their predictors covering at least one TF for the enriched motifs (37.8%/9.9% = 3.8-fold enrichment). Clustering DHSs into 2000 or 5000 pathways yielded similar results (**Supplementary Fig. 15d-f**). Since we used clusters of co-expressed genes as predictors, having a TF in the predictor implies that genes co-expressed with the TF, which often come from related biological processes or pathways, are also included in the predictors. **Supplementary Figure 15g-j** and **Supplementary Table 2** show a sample DHS pathway which is active in human skeletal muscle myoblasts (HSMM) and skeletal muscle myotubes differentiated from HSMM (HSMMtube) (**Methods**). TFs known to be involved in muscle development such as MYF6, MYOD1 and MYOG were found in the pathway’s predictor genes (**Supplementary Figure 15i**), and motifs of these TFs were among the top enriched motifs in the DNA sequences of DHSs in this pathway (**Supplementary Figure 15h**). Gene Ontology (GO) analysis of the predictor genes further identified “muscle cell development” and “skeletal muscle tissue development” as enriched GO terms for this pathway (**Supplementary Figure 15j**). Thus, predictor

genes of this pathway were enriched in biological processes consistent with the cell types in which the DHS pathway is active. The same analyses were also conducted on other pathways. The complete catalog of BIRD-selected predictor genes for each DHS and DHS pathway, and the enriched DNA motifs and GO terms for each pathway are provided at <https://zhiji.shinyapps.io/CABS/> as an online resource.

These results are now added to the revised manuscript (new section "**Predictors selected by BIRD**" and **Supplementary Fig. 15**).

The comparison of BIRD with ChromImpute is not entirely convincing, as in the plot in the main figure Pearson correlation of P-T values for BIRD, which are DNase levels, are compared with the correlation of P-T values for ChromImpute, which are predicted log₁₀ p-values. The authors address this in the supplementary figures by using Spearman correlation, which may level the playing field somewhat, but while they state that "neither method consistently outperformed the other", there are several outlier cases (GM12878, Monocytes and NHEK) for which ChromImpute performs well above BIRD, and these are also the cell types with the largest deviation from the average (based on the average prediction model). The result is still important, however, given that ChromImpute is based on histone modification which may be expected to better predict DH, since H3K27ac and H3K4me3 mark active enhancers and promoters.

Thank you for raising this point which we will address below.

First, we have checked the three outlier cases GM12878, Monocyte and NHEK for which ChromImpute performs well above BIRD. We found that this was because ChromImpute used more information in the prediction. ChromImpute used observed data from 127 epigenomes to train prediction models for imputing missing data in these epigenomes. Thus, its prediction was based on more cell types than what we had in the BIRD analysis. In BIRD, we only used 57 cell types with both DNase-seq and gene expression data for our analysis, and we used leave-one-out approach for testing, which means for each test cell type we only used 56 cell types for training the model. For the three outlier cases, the ChromImpute training data (after excluding these three test cell types) contained highly similar cell types "Primary monocytes from peripheral blood", "Primary B cells from peripheral blood", and "Foreskin Keratinocyte Primary Cells" (similar to Monocyte, GM12878, and NHEK respectively). By contrast, the 57 cell types used in the BIRD analysis only contained one Monocyte cell line (**Supplementary Table 1**). When it was used as the test cell type in the leave-one-out cross-validation, no other monocyte samples were contained in the training data. Similarly, GM12878 was the only lymphoblastoid cell line, and NHEK was the only epidermal keratinocytes cell line in our 57 cell types. Thus, BIRD predictions for these three cases were based on models trained using much less similar cell types. This is now explained in the Methods section "**Comparing BIRD with ChromImpute**". We also note that ideally, we would like to conduct a comparison in which both BIRD and ChromImpute are trained using the same cell types. However, we found this difficult to do because (1) the cell types analyzed in ENCODE (used for BIRD) and those in Epigenome Roadmap (used for ChromImpute) were different, and (2) training ChromImpute requires us to download and process all types of epigenomic data from Epigenome Roadmap project which is beyond the storage and computing capacity available to us. Fortunately, even though we cannot conduct such an ideal comparison, our results show that BIRD trained using fewer cell types and fewer data types (gene expression alone) already performed comparably with ChromImpute which were trained based on more cell types and more data types.

Second, we appreciate that this reviewer acknowledges that “The result is still important, however, given that ChromImpute is based on histone modification which may be expected to better predict DH, since H3K27ac and H3K4me3 mark active enhancers and promoters.”

Finally, in response to reviewer 1’s comments, we have now conducted further comparison between BIRD and ChromImpute in terms of their ability to predict differential DH between two cell types. This analysis shows that BIRD outperformed ChromImpute substantially for comparing highly similar cell types, and ChromImpute performed slightly better than BIRD for comparing cell types with lower similarity levels (**Fig. 2j, Supplementary Fig. 14e-f**). These results provide more evidence to demonstrate the relative merits of BIRD.

Prediction of TFBS using expression data is interesting, however the authors could surely find a better example than MYC - as they say, the motif is bound by many other transcription factors, and predictions at an FDR of approximately 50% are not generally useful.

Thank you for the advice. Since the MYC example is confusing, we have replaced it with another example ELF1. For this new example, the sensitivity at 25% and 50% FDR level was 0.64 and 0.88 respectively. At these FDR levels, BIRD generated 9000 and 19000 predicted ELF1 binding sites.

We also note that in practice FDR can be reduced by sacrificing some sensitivity. For example, if one wants cleaner predictions, one may reduce FDR to 10%. At this FDR level, the sensitivity is 0.43, and one still has 5000 predicted binding sites. Thus, one can still make a lot of new discoveries compared to not using BIRD to make predictions at all.

Finally, an important point is that the TFBS prediction performance presented in this article does not represent the upper limit one can achieve using BIRD. TFBS prediction is a complex problem involving integration of multiple types of information such as chromatin accessibility and DNA motif, etc. Our analyses here were focused on investigating different ways to use one information type (i.e., chromatin accessibility). There is still plenty of room for improving TFBS prediction accuracy by optimizing the use of other information types. For instance, by using a more sophisticated motif model that accounts for the intra-motif correlation (i.e., correlation among different positions within a motif), we could increase the sensitivity of BIRD prediction for ELF1 at the 10%, 25% and 50% FDR level from 0.43, 0.64 and 0.88 to 0.51, 0.75 and 0.88. The sensitivity at 10% and 25% FDR level outperformed the prediction based on true DNase-seq coupled with the conventional motif model which ignores intra-motif correlation (**Supplementary Notes, Supplementary Fig. 28**). These results imply that using more sophisticated motif models may allow one to reduce the FDR while keeping sensitivity unchanged. Besides intra-motif correlation, information such as phylogenetic conservation, motifs of collaborating TFs, and genomic locations relative to genes may also improve TFBS prediction. However, the focus of this article is to illustrate the use of predicted DH in the absence of experimental regulome data rather than building the optimal TFBS prediction pipeline. As such, systematically exploring the optimal use of all these non-DH predictors in TFBS prediction is beyond the scope of this study. For this reason, the significance of our results cannot be evaluated just based on the presented prediction accuracy. The most significant and useful thing we contributed through our TFBS analysis is our demonstration of the feasibility of using transcriptome to predict TFBSs when experimental regulome data is not available. This adds a new tool to the toolbox one can use to build better TFBS prediction pipelines in the future.

The resource provided which has reanalysed 2000 GEO samples could be useful to some researchers, however it seems odd that using the website more results can be submitted using the text field than

as BED coordinates ("2MB size limit. Larger entries can be entered in text area below"). Manually selecting and pasting in large queries is difficult and makes automation impossible on most systems.

This is a good point. We have now updated the website to remove the 2MB size limit of input BED files.

The demonstrate that BIRD can sometimes improve DNase predictions based on poor replicates, but a more interesting use case would be the detection of outlier samples, as is done in the ChromImpute paper, and a similar analysis would be more convincing here in demonstrating its usefulness as a QC metric.

Thank you for raising this point. As you suggested, we have tried to use BIRD predictions as pseudo-replicate for quality assessment. Our analyses indicate that the correlation between BIRD-predicted DH and experimentally measured DH by DNase-seq (BIRDcor) can be used as a QC metric. We compared this metric to a number of existing QC metrics currently used by ENCODE. BIRDcor was found to perform favorably. These results are now presented in **Supplementary Notes S5 "Using predictions for data quality check"** and **Supplementary Figures 34-38**.

Of note, in our analyses, we found that different QC metrics are complementary to each other. Poor-quality samples failed to be identified by one QC metric may be identified by other QC metrics. Thus, while BIRDcor can provide a useful metric for QC and performs favorably compared to existing QC metrics, in practice we recommend using multiple QC metrics together. This is stated in the **Supplementary Notes** as:

"Supplementary Figure 38 also shows that different QC metrics are complementary to each other. Poor-quality samples failed to be identified by one QC metric may be identified by others. For instance, Chorion_Rep2 did not look problematic based on seq-depth, SPOT, NSC and RSC, but it was identified by BIRDcor and PBC as a sample with relatively low quality. Some samples did not look problematic based on BIRDcor, but they could be ranked low by other QC metrics. Thus, in practice, we recommend using multiple QC metrics together. These QC metrics often examine samples' quality from different perspectives and use different types of information. Therefore, one would be more confident about the judgement of poor quality if a sample's poor quality is indicated by multiple QC metrics. In this regard, BIRDcor provides an additional piece of information one can use for QC check, and it contributes a new metric to the existing QC toolbox."

The absence of evidence that the method works comparably well on RNA-seq as on exon array expression data is a quite serious omission, since most expression data in the future will come from RNA-seq and related methods and not exon arrays.

This is an excellent point. Indeed, we have tested BIRD using RNA-seq data. For example, we used leave-one-cell-type-out cross-validation to train and test BIRD on 70 RNA-seq and matching DNase-seq samples obtained from the Roadmap Epigenomics project. The conclusions from this analysis are essentially the same as the conclusions we obtained from exon array data in this article. For example, BIRD also significantly outperformed the random model, and the prediction accuracy was similar to the accuracy observed in the exon array data (see figure shown below).

Figure. BIRD predicts DH using RNA-seq. The plots show prediction performance of BIRD and random prediction models ("BIRD-permute") in leave-one-cell-type-out cross-validation. **(a)** Distribution and mean of cross-locus correlation r_L from all samples. **(b)** Distribution and mean of cross-sample correlation r_C from all loci. **(c)** Squared prediction error (τ).

More importantly, we have also found that BIRD can be used to make predictions using RNA-seq data from samples with small number of cells and single-cell RNA-seq data. This is important since conventional high-throughput technologies for mapping regulatory element activities such as ChIP-seq, DNase-seq and FAIRE-seq cannot analyze samples with small number of cells. The recently developed ATAC-seq allows regulome mapping in small-cell-number samples, but its signal in single cell or samples with ≤ 500 cells remains discrete or noisy. Compared to these technologies, measuring transcriptome by RNA-seq in single-cell and small-cell-number samples is more mature. Our results show that one can globally predict chromatin accessibility and infer regulome using RNA-seq. Genome-wide chromatin accessibility predicted by RNA-seq from 30 cells is comparable with ATAC-seq from 500 cells. Predictions based on single-cell RNA-seq (scRNA-seq) can more accurately reconstruct bulk chromatin accessibility than using single-cell ATAC-seq (scATAC-seq) by pooling the same number of cells. Integrating ATAC-seq with predictions from RNA-seq increases power of both methods. Thus, transcriptome-based prediction can provide a new tool for decoding gene regulatory programs in small-cell-number samples. All these results demonstrate the value of BIRD.

These results can be found in a bioRxiv preprint: Zhou, W., Ji, Z. & Ji, H. *Global Prediction of Chromatin Accessibility Using RNA-seq from Small Number of Cells*. bioRxiv, 035816.

(<http://www.biorxiv.org/content/early/2016/01/01/035816>) which we are submitting for publication as an independent article. As you can see, we actually have done extensive amounts of work to test

BIRD in RNA-seq. The substantial results and findings there require an independent article to elaborate and discuss. Thus, they are not included as part of this current manuscript.

The relationship between our current manuscript and this other manuscript is that our current manuscript is focused on (1) establishing the feasibility of using gene expression to predict DHS, and (2) establishing the BIRD method. Thus it is the original BIRD method paper. By contrast, the objective of the other manuscript is to demonstrate that the BIRD method established in our current manuscript can also be applied to RNA-seq, and more importantly it can also be applied to small-cell-number and single-cell RNA-seq data to produce better signals than existing experimental technologies for mapping regulomes in small-cell-number and single-cell samples (e.g. single-cell ATAC-seq).

Reviewers' comments:**Reviewer #1 (Remarks to the Author):**

The authors have made many important improvements and additions to this manuscript. The authors have demonstrated the applicability of BIRD in multiple new ways, including applying it to a neuronal differentiation dataset and using it to predict which transcription factors are binding in the differentially open regions. The authors have also added multiple analyses to interpret the model. In addition, they have added comparisons to existing methods. There are some additional more minor ways that the authors could improve the manuscript, and I have described them in the attached pdf.

Reviewer #2 (Remarks to the Author):

The authors' response and revisions are extensive and thoughtfully done. All of my concerns from the initial review have been addressed in a satisfactory manner.

Reviewer #3 (Remarks to the Author):

The authors have responded thoroughly and adequately to my original concerns and questions.

Purple: Reviewer's Questions and Comments

BLACK: Responses

RED: Revised texts in the article

Reviewer #1 (Remarks to the Author):

The authors have made many important improvements and additions to this manuscript. The authors have demonstrated the applicability of BIRD in multiple new ways, including applying it to a neuronal differentiation dataset and using it to predict which transcription factors are binding in the differentially open regions. The authors have also added multiple analyses to interpret the model. In addition, they have added comparisons to existing methods. There are some additional more minor ways that the authors could improve the manuscript, and I have described them below.

Thank you for carefully reviewing our revised manuscript. We have revised our manuscript according to your suggestions which has further improved our manuscript. Please see our detailed responses below.

The additional comparisons are very helpful. They demonstrate that the method in this paper is much faster than methods other than KNN and gives a lower prediction error than KNN. However, they were done using only 1% of the genome, and it is possible that some other methods would work better with more available data. To be fully convincing, the authors would need to show that their method gives lower prediction error than KNN on a larger percentage of the genome.

Thank you for your suggestion. We have added a whole genome (i.e., all DHSs) prediction performance comparison between KNN and $BIR(\bar{X}, Y)$ (see **Supplementary Notes S1** and **Supplementary Fig. 6**). The results show that $BIR(\bar{X}, Y)$ had higher prediction accuracy than KNN which is consistent with our results from 1% of the genome. We would like to point out that this is expected because the 1% of the genome used for comparing different prediction methods were randomly sampled from all DHSs. Based on the statistical sampling theories, a population characteristic can be unbiasedly estimated using a random sample representative of the population. Thus, the performance of different prediction methods on these 1% DHSs should represent the performance of the methods on all DHSs. Accordingly, we have added the following paragraph to **Supplementary Notes S1**:

*“Since KNN and $BIR(\bar{X}, Y)$ had comparable computational efficiency, we further compared their prediction accuracy using all 912,886 DHSs rather than 1% of random DHSs. To this end, we trained both KNN and $BIR(\bar{X}, Y)$ using the 40 training cell types, with optimal parameters determined by five-fold cross-validation using the training data. The trained models were then applied to predict DH in the 17 test cell types. The prediction accuracy is shown in **Supplementary Figure 6**. Consistent with the results obtained using 1% of random DHSs, $BIR(\bar{X}, Y)$ showed higher prediction accuracy than KNN in this whole-genome comparison. This is not surprising because based on the*

statistical sampling theories, a population characteristic can be unbiasedly estimated using a random sample representative of the population.”

The subsampling analysis is very helpful, though I think that the y-axis for Supplementary Figure 39b should say “Fraction” instead of “Percentage.” In addition, I appreciate the clarification in the Discussion section explaining that the method cannot identify novel open chromatin regions but can predict which subset of open chromatin regions from other cell types are likely to be active in a new cell type that has available gene expression data.

Thank you for the comments and suggestion. We have now changed “Percentage” to “Fraction” as suggested (see **Supplementary Figure 43b** in the revised manuscript which was **Supplementary Figure 39b** in the old manuscript).

[Re: Cross-cell-type prediction accuracy varies greatly among different loci and factors that influence cross-cell-type prediction accuracy]

These additions substantially improve the manuscript and enhance my understanding of when the method works best.

However, I would not have removed all of the “noisy” loci when doing these comparisons. I would not expect the method to do well on loci that are active in every training cell type and inactive in every test cell type, so I think that removing those loci is reasonable. However, I would expect the method to do well on loci that are active in only a subset of the training cell types, so I would not remove those.

In addition, I did not understand the difference between Figure 2 a-d and Supplementary Figure 9. It would be great if the authors could clarify this.

Thank you for your comments.

Regarding the removal of “noisy loci”:

We would like to clarify that your statement that “I would expect the method to do well on loci that are active in only a subset of the training cell types, so I would not remove those” is not completely correct. For predicting cross-cell-type biological variation, the correlation between the predicted and true DH levels not only depends on whether the training data are informative for training a good prediction model but also depends on whether the test data contain real biological variation.

Before we elaborate this, it is useful to keep four key facts in mind which provide the context for our discussion.

1. Recall that there are two types of DNase-seq data: training data and test data.
2. All loci analyzed in this study are active in at least one training cell type. This is because our data preprocessing has filtered out loci that are inactive in all training cell types.
3. Among the analyzed loci, there is a subset of loci that are inactive in all test cell types. These loci are defined by us as “noisy loci”. They were not filtered out in our preprocessing step because one cannot assume that DNase-seq data from test cell types are available in real applications of BIRD. Based on this definition,

- “noisy loci” are a subset of all loci analyzed by BIRD, and all “noisy loci” are active in at least one training cell type but they are inactive in all test cell types.
4. The definition of “noisy loci” is dependent on the test cell types. Assuming that the training data are fixed, changing the test cell types will change the set of loci labeled as “noisy loci”. Similarly, in real applications, “noisy loci” are defined based on the cell types on which predictions are made. These cell types vary from one application to another. Therefore, the definition of “noisy loci” also varies from one application to another application.

For noisy loci, data in test cell types are primarily noise. Statistically, the correlation between a random noise and an uncorrelated variable is always zero. Therefore the cross-cell-type correlation between the predicted and observed DH levels at these noisy loci is expected to be low in test cell types. This low correlation does not depend on whether a noisy locus is active in some or in all training cell types.

To help understand it mathematically, let random variable $Y = S + e$ be the observed DH level in a test cell type. Here S represents the underlying true DH level in the test cell type and e represents independent random noise. Both S and e are unobserved. The biological signal S may be predictable using gene expression data, but the random noise e is unpredictable. Let \hat{Y} be the predicted DH level. Our evaluation of the cross-cell-type prediction accuracy is based on studying the correlation between \hat{Y} and Y across test cell types. For a “non-noisy locus”, the covariance between \hat{Y} and Y is $Cov(\hat{Y}, Y) = Cov(\hat{Y}, S + e) = Cov(\hat{Y}, S) + Cov(\hat{Y}, e) = Cov(\hat{Y}, S)$. Here, $Cov(\hat{Y}, e) = 0$ because \hat{Y} represents predictions based on gene expression data and e is independent random noise. As a result, the covariance $Cov(\hat{Y}, Y) = Cov(\hat{Y}, S)$ reflects the ability of the prediction model to capture the variation of the biological signal S across cell types, and it is often non-zero. For a “noisy locus”, the real biological signal $S = 0$ in all test cell types, and therefore $Cov(\hat{Y}, Y) = Cov(\hat{Y}, e) = 0$. As a result, the correlation between \hat{Y} and Y is zero. This result does not depend on the activity status of the noisy locus in the training data. Even if the locus is active in a subset of training cell types and one can obtain a very good prediction model, this covariance is still zero in test cell types.

The low cross-cell-type correlation observed in test cell types at a noisy locus does not necessarily imply that the prediction model itself is inaccurate. It simply reflects the fact that the test data do not contain any meaningful biological variation other than random noise. If one could change the test data to include some new test cell types in which the locus is active, then the “noisy locus” would become a “non-noisy” locus and this could change the covariance $Cov(\hat{Y}, Y)$ in test cell types to a very different value. From this perspective, the reviewer is partially correct in stating that the prediction models for loci that are active in only a subset of the training cell types should be good models. However, the reviewer missed an important point that the good performance of these models can only be observed when the test data are not all noise. For a noisy locus, whether a model is good or not in terms of predicting cross-cell-type variation is not observable in the test cell types. Thus, including noisy loci in the analysis will produce misleading results regarding models’ cross-cell-type prediction performance. Moreover, for a noisy locus, whether the model is good or not in terms of predicting cross-cell-type variation becomes irrelevant because all variation at such a locus is noise and studying differential biological signal is therefore meaningless. From these perspectives, these noisy loci should be removed from the analysis.

In real applications, “noisy loci” will be defined based on the cell types on which one wants to make predictions. Thus, the set of loci labeled as “noisy loci” varies from one application to another. As a result, the loci removed from the analysis will be different in different studies. For a locus that is active in a subset of training cell types and therefore has a good prediction model, the locus may be filtered out as “noisy locus” in some applications where the test cell types are all noise and predicting cross-cell-type variation is not meaningful, and it may be retained as “non-noisy locus” in other applications where the test cell types contain some real biological signals and there is a need for predicting cross-cell-type variation. In both scenarios, no crucial information is lost.

In summary, one should not expect the prediction model to produce high cross-cell-type correlation r_c in test cell types at the noisy loci. Although a low r_c at a noisy locus does not necessarily imply that the prediction model is not good, the cross-cell-type prediction performance of the model at such a locus becomes irrelevant. In practice, it is not useful for predicting how biological signals change across a set of new cell types at a locus when the locus does not have any real biological signal in these new cell types.

In the revised manuscript, we added the following texts in the subsection “*Cross-cell-type prediction accuracy varies greatly among different loci*” to clarify these:

“For these loci, the true DH levels are essentially noise, and the cross-cell-type correlation between the predicted DH levels and random noise is expected to be low. Therefore, these noisy loci are not informative for evaluating the performance of predicting biological variation across cell type.”

In the **Online Method** section “**Factors affecting cross-cell-type prediction accuracy**”, we further added:

“Conceptually, the low r_c at a noisy locus does not necessarily imply that the prediction model for that locus is inaccurate. It simply reflects the fact that the data in test cell types are primarily noise and do not contain enough biological variation for evaluating the model’s ability to predict variation of biological signals across cell types. It is possible that a locus is active in a subset of training cell types and has a very good prediction model. However, the good cross-cell-type prediction performance of a model can only be observed when the test data are not all noise. When the test data are all noise, whether the model is good or not in terms of predicting cross-cell-type biological variation is not observable. Moreover, in real applications, it is not useful to study the cross-cell-type variation or differential DH of a locus in a set of new cell types if DH levels at this locus in these new cell types are all noise. For these reasons, we separated “noisy loci” from “non-noisy loci” in our subsequent analysis and evaluation.”

“We note that the definition of noisy loci and test data factors (i.e., loci with low max-min spread, low CV or high cell-type-specificity in test data) are dependent on the test cell types. In real applications, they will be defined based on the cell types on which BIRD is applied to make predictions. For simplicity, these cell types are also called test cell types below. Different applications may involve different test cell types. Thus, the set of loci labeled as noisy loci, low max-min spread, low CV or high cell-type-specificity in test cell types varies from one application to another. As a result, the loci removed from the analysis by the above filters will be different in different studies. For example, consider a locus that is active in a subset of training cell types and therefore has a good prediction

model. The locus may be filtered out as a “noisy locus” in some applications where the data in test cell types are all noise and predicting cross-cell-type variation at the locus is not practically useful, and it may be retained as “non-noisy locus” in other applications where the data in test cell types contain some real biological variation and there is a need for studying differential DH at the locus.”

Regarding the difference between Figure 2 a-d and Supplementary Figure 9:

Recall that we have both training data and test data. Therefore genomic loci can be categorized either based on DH levels in the training cell types or based on DH levels in the test cell types. Both **Figure 2a-d** and **Supplementary Figure 10** in the revised manuscript (i.e., the **Supplementary Figure 9** in the old manuscript) are used to study how different factors in the training data and test data influence the cross-cell-type correlation r_c . The difference between these two figures lies in how the test data factors are defined. In **Figure 2a-d**, test data factors are defined using the true DH levels obtained from the real DNase-seq data. Here, loci are grouped into different categories based on their DH values in real DNase-seq data. In **Supplementary Figure 10**, test data factors (including noisy loci, max-min spread, coefficient of variation, and cell type specificity) are defined using the BIRD predicted DH levels. In other words, loci are categorized based on their predicted DH values generated by BIRD from gene expression data.

In **Figure 2a-d**, our purpose is to understand how different training and test data factors influence the cross-cell-type prediction accuracy. Therefore, here we used the true DNase-seq data to categorize loci.

In real applications, however, BIRD is most useful when DNase-seq data for new cell types are not available. Therefore one cannot categorize loci based on the real DNase-seq data. However, one may still want to identify loci for which the cross-cell-type correlation r_c is potentially high. For this purpose, one can categorize loci based on the BIRD predicted DH levels. In **Supplementary Figure 10**, we conducted an analysis similar to **Figure 2a-d** but used BIRD predicted DH instead of the true DNase-seq data in the test cell types to categorize loci. The purpose of **Supplementary Figure 10** is to verify that BIRD predicted DH in test cell types can be used to screen for loci with potentially accurate cross-cell-type predictions.

In the revised manuscript, we added the following texts to clarify these:

*“By examining the true DH levels measured by DNase-seq in the training and test cell types, it was found that multiple factors may influence cross-cell-type prediction accuracy of a locus. ... Together, these factors and noisy loci explained the majority (85%) of loci with low r_c (i.e., $r_c < 0.25$) (**Fig. 2b-d**). In real applications, BIRD is most useful for making predictions in new cell types for which DNase-seq data are not available. Therefore, we repeated this analysis by using BIRD-predicted DH levels (instead of true DH levels) in the test cell types and the true DH levels in training cell types for locus stratification. The analysis produced similar results (**Supplementary Fig. 10**).”*

[Re: comparison with PIQ and CENTIPEDE] These additional analyses are very helpful. While the authors have provided a good explanation of why they require the presence of a motif for a TF in order to predict binding, I think that the authors should directly state that this is a limitation of their method (and of many others)

Thank you for your suggestion. We have now added a few sentences in the manuscript to discuss this limitation. Please see the section **“Predicting Transcription Factor Binding Sites”**:

“One limitation of this approach is that it requires accurate motif information and the prediction is contingent on the presence of TF binding motifs. This limitation, however, is not unique to BIRD. It is common to methods such as PIQ and CENTIPEDE that use chromatin accessibility to predict TFBSs.”

[Re: differential analysis] These additional studies are great additions to the paper. I would recommend adding something about the findings from the differential regulatory signal analysis to the introduction because I think this will make the paper more interesting to some readers.

Thank you for your suggestion. We have now added a few sentences in the introduction section to mention the findings from differential analysis:

“We show that BIRD can provide practically useful predictions of chromatin accessibility using gene expression. BIRD-predicted DH can be used to predict transcription factor binding sites (TFBSs), turn publicly available gene expression samples in GEO into a regulome database, and serve as pseudo-replicates to facilitate regulome data analyses. It can also be used to predict differential regulatory element activities such as changes of chromatin accessibility between different cell types or differentiation time points.”

For the comparison to ChromImpute, it would be great if the authors could add a comparison of Spearman correlations in addition to Pearson correlations.

Thank you for your suggestion. We have added a comparison of Spearman’s rank correlation between ChromImpute and BIRD (see **Supplementary Fig. 15 a,d,g**). Replacing the Pearson’s correlation with the Spearman’s rank correlation did not change our conclusions. We updated the **Online Methods** section **“Comparing BIRD with ChromImpute”** accordingly:

*“Second, we replaced the Pearson’s correlation by Spearman’s rank correlation and repeated the above analysis again (**Supplementary Fig. 15a**).”*

*“Similarly, we have also calculated the cross-cell-type correlation r_C using these **four** types of analyses. The results were shown in **Figure 2i** and **Supplementary Figure 15d-f**.”*

*“The results were shown in **Figure 2j** and **Supplementary Figure 15g-i**. **Figure 2j** shows Pearson’s correlation between the predicted and true differential signals of the*

same type (i.e., read-signal vs. read-signal, or pval-signal vs. pval-signal) across all loci and differential loci. **Supplementary Figure 15g** shows Spearman's rank correlation between the predicted and true differential signals of the same type."

For the comparison to ChIP-qPCR for H3K4me1, it would be great if the authors could report the Spearman correlation in addition to the Pearson correlation.

Thank you for your suggestion. We have added Spearman's rank correlation in the main manuscript. The Spearman's rank correlation is similar to the Pearson's correlation.

"The predicted DH difference correlated well with the ChIP-qPCR measured H3K4me1 difference (Fig. 6c, Pearson's correlation=0.67, Spearman's rank correlation=0.61)."

For the Supplementary Table 10, it would be great if the authors could report p-values in addition to fold-changes.

Thank you for your suggestion. We have included the adjusted p-values (i.e., FDR) in Supplementary Table 10. The **Online Methods** section "**Predicting differential DH during neuron differentiation**" is updated accordingly:

"For each motif, statistical significance of the enrichment was evaluated using one-sided Fisher's exact test (using the numbers of motif sites and the numbers of non-repeat base pairs in differential DHSs and control regions). P-values were adjusted using the BH procedure to obtain FDR. Motifs with FDR < 0.05 and enrichment level >2 and with at least 50 motif sites in differential DHS regions were reported as enriched motifs (Supplementary Table 10)."

[Re: TFBS prediction] These explanations and additions are helpful. It would be great if the author's could put the MYC results back into the manuscript (in addition to the ELF1 results) and explain why the results for MYC are not expected to be good.

Thank you for your suggestion. Taking your advice, we have now put back the MYC TFBS prediction results in Supplementary materials (see **Supplementary Notes S4** and **Supplementary Fig. 29**). The analysis and plots were updated to be consistent with the analysis of other TFs. This dataset does not have replicate samples and therefore one cannot use the IDR procedure to define peaks. ChIP-seq peaks were identified using CisGenome two-sample (IP vs. control) peak calling at the FDR cutoff of 0.01.

We explained the potential complications of interpreting the results:

"We note that accurate prediction of MYC binding sites can be complicated by the fact that the motif bound by MYC, known as E-box motif, is also recognized by many other TFs (e.g., USF1). Thus, motif sites with high predicted DH levels can be bona fide binding sites of other TFs but labeled as false positives by MYC ChIP-seq data. For example, at 80% sensitivity level, the FDR of BIRD was 34%. The 34% predictions

classified as false positives may be real binding sites of other E-box binding TFs. Considering this complication, BIRD performed reasonably well."

Note that our old presentation of the MYC example in the original manuscript did not provide the sensitivity-FDR curve, and we only discussed predictions at one peak calling cutoff which corresponds to only one data point in the sensitivity-FDR curve. At that cutoff, the FDR was not very low and that caused the confusion. As we elaborated in our first-round revision (R1), looking at only one data point in the sensitivity-FDR curve does not provide the whole picture since FDR is adjustable and one can get smaller FDR at the cost of losing some sensitivity. In this current revision (R2), we added the whole sensitivity-FDR curve (**Supplementary Figure 29a**), and one can see that the prediction performance was not bad, particularly when BIRD is compared with the mean DH and motif only methods.

These additions are helpful. In Supplementary Figure 15, I think that the y-axis should say "Fractions" and not "%."

Also, it would be great if the authors could add p-values in addition to fold-changes in Supplementary Table 2.

Thank you for your suggestions. We have changed "%" and "percentage" to "fraction" in the figure as suggested (see **Supplementary Figure 16** in the revised manuscript which was **Supplementary Figure 15** in the old manuscript). We have also added adjusted p-values (FDR) to **Supplementary Table 2**.

The **Online Methods** section "**Analysis of predictors selected by BIRD**" is updated accordingly:

"For each motif, to test whether the motif is significantly enriched in the target regions as compared to the control regions, one-sided Fisher's exact test was applied (using the numbers of motif sites and the numbers of non-repeat base pairs in target regions and control regions). P-values were adjusted using Benjamini-Hochberg (BH) procedure to obtain FDR. Motifs that showed FDR < 0.05 and enrichment ratio larger than 2 and had more than 50 motif sites in target regions were labeled as enriched motifs."

"For each DHS pathway, we annotated it with cell types in which it is active. In order to do so, we first applied one-sided Wilcoxon rank-sum test to evaluate whether the mean DH within the DHS pathway is different from the mean DH value of all other DHSs in each cell type. P-values from all cell types were then adjusted using the BH procedure to obtain FDR. For each cell type, we also calculated the DH enrichment of a DHS pathway as the difference (δ) between the average DH value (at log2 scale) within the DHS pathway and the average DH value from all DHSs. According to the FDR and DH enrichment, cell types with FDR < 0.05 and $\delta > 1$ were considered as active cell types for each DHS pathway."

[Re: use BIRD for quality assessment] These additional analyses are very helpful. For Figures 37-38, it would be great if true positive rate versus false discovery rate curves were added.

Thank you for your suggestions. We have now added true positive rate versus false discovery rate curves (i.e., sensitivity-FDR curve) for the comparison of BIRDcor with ENCODE QC metric (see **Supplementary Figs. 37, 38(d-f), 40, 41(d-f)**). The results are consistent with what we observed from the ROC curves and are discussed in **Supplementary Notes S6**:

*“Using these poor-quality samples as gold standard, we compared BIRDcor and the tested ENCODE QC metric in terms of their ability to detect poor-quality samples by computing their receiver operating characteristics (ROC) and **sensitivity versus FDR curves (Supplementary Figs. 36-37)**. The results show that BIRDcor had better performance than the ENCODE QC metrics in most of the comparisons. **We note that in the sensitivity versus FDR comparisons, all methods had relatively low sensitivity when the FDR was low (Supplementary Fig. 37)**. This is not surprising because different metrics are complementary to each other and each QC metric can only detect a subset of low quality samples while missing many low quality samples identified by other metrics.”*

*“Next, we compared all methods together. ... Regardless of the value of x , BIRDcor showed comparable performance to the other QC metrics in terms of both ROC and sensitivity-FDR curve (ROC: **Supplementary Fig. 38a-c**; **Sensitivity-FDR: Supplementary Fig. 38d-f)**.”*

*“Based on the new gold standard definitions, BIRDcor still performed comparable to or better than the other QC metrics (**Supplementary Figs. 39, 40, 41**).”*

REVIEWERS' COMMENTS:

Reviewer #1 (Remarks to the Author):

The authors have addressed all comments and concerns.